# Loss-of-function mutations in Keratin 32 gene disrupt skin immune homeostasis in pityriasis rubra pilaris

Peidian Shi [1,2,8], Wenjie Chen [1,2,8], Xinxing Lyu[1,2], Zhenzhen Wang [1,2], Wenchao Li[1,2], Fengming Jia[1,2], Chunzhi Zheng[1,2], Tingting Liu[1,2], Chuan Wang[1,2], Yuan Zhang[1,2], Zihao Mi[1,2], Yonghu Sun [1,2], Xuechao Chen[1,2], Shengli Chen[1,2], Guizhi Zhou[1,2], Yongxia Liu[1,2], Yingjie Lin[1,2], Fuxiang Bai[1,2], Qing Sun[3], Monday O. Ogese [4], Qiang Yu [5], Jianjun Liu [5], Hong Liu [1,2,6] ✉ & Furen Zhang [1,2,6,7] ✉

Pityriasis rubra pilaris (PRP) is an inflammatory papulosquamous dermatosis, characterized by hyperkeratotic follicular papules and erythematous desquamative plaques. The precise pathogenic mechanism underlying PRP remains incompletely understood. Herein, we conduct a case-control study involving a cohort of 102 patients with sporadic PRP and 800 healthy controls of Han Chinese population and identify significant associations ($P = 1.73 \times 10^{-6}$) between PRP and heterozygous mutations in the Keratin 32 gene (*KRT32*). KRT32 is found to be predominantly localized in basal keratinocytes and exhibits an inhibitory effect on skin inflammation by antagonizing the NF-κB pathway. Mechanistically, KRT32 binds to NEMO, promoting excessive K48-linked polyubiquitination and NEMO degradation, which hinders IKK complex formation. Conversely, loss-of-function mutations in *KRT32* among PRP patients result in NF-κB hyperactivation. Importantly, *Krt32* knockout mice exhibit a PRP-like dermatitis phenotype, suggesting compromised anti-inflammatory function of keratinocytes in response to external pro-inflammatory stimuli. This study proposes a role for KRT32 in regulating inflammatory immune responses, with damaging variants in *KRT32* being an important driver in PRP development. These findings offer insights into the regulation of skin immune homeostasis by keratin and open up the possibility of using KRT32 as a therapeutic target for PRP.

Pityriasis rubra pilaris (PRP) is a chronic immune-mediated papulosquamous dermatosis characterized by follicular keratotic papules, orange-red scaly plaques, and palmoplantar keratosis; this disorder was initially described as a variant of psoriasis in 1835[1] and then designated "pityriasis pilaris" in 1856[2]. Both genders are equally prone to develop PRP[3]; The estimated prevalence of PRP is 1 in 5000 patients presenting with skin disease in Great Britain[4] or 1 in 50,000 in India[5], with the first peak in the first decade of life and the second peak in the sixth decade of life[4]. The pathogenesis of PRP is primarily attributed to the dysregulation of immune function of epidermal keratinocytes, with only sparse lymphocytic infiltration observed around superficial dermal blood vessels in the uppermost dermis[6,7]. Dysfunctional keratinocytes potentially act as a pivotal driver to promote the development of PRP.

Although keratinocytes play a pivotal role in maintaining the biochemical and physical integrity of the skin[8], it has become evident that keratinocytes actively participate in immune responses, extending their role beyond structural functions over the past decades[9,10]. In particular, keratinocytes have been recognized as critical contributors in both initiation and maintenance phases of papulosquamous dermatosis, including psoriasis, actively participating in immune processes through multiple mechanisms and regulatory factors[11,12]. In addition, keratin-related genes have also been implicated in immune responses, such as KRT1 regulates innate immunity by controlling IL-18 release[13] and abnormal expression of keratin genes like KRT6, KRT16, and KRT17 is observed in psoriatic lesions, leading to increased keratinocyte proliferation, and heightened inflammation[14–19]. In the case of PRP, a disorder characterized by keratinocyte proliferation, mutations in the CARD14 gene, which demonstrate tissue-specific expression in the skin, are associated with familial cases of PRP[7,20,21]. As an activator of the NF-κB signaling pathway, CARD14 mutations can induce the activation of the NF-κB pathway in keratinocytes, ultimately leading to skin inflammation[7]. Interestingly, recent research has discovered that patients with PRP carrying CARD14 mutations may undergo spontaneous repair of these mutations in keratinocytes through homologous recombination, resulting in the restoration of normalcy in the affected skin, although these mutations still persist in the dermis and peripheral blood, underscoring the critical role of keratinocytes in the pathogenesis of PRP[22]. Furthermore, CARD14 mutations have mainly been reported in familial PRP cases, which only account for 5% of all cases[4,7,23]. Therefore, further investigation is necessary to identify additional genes with keratinocyte-specific expression that can potentially regulate skin immune homeostasis and contribute to the pathogenesis of PRP.

In this work, to comprehensively investigate the genetic and functional aspects underlying PRP, especially the involvement of specific expression of keratinocyte genes, we perform a whole exome sequencing (WES). Our findings reveal a significant association between PRP and the Keratin 32 gene (KRT32), which exerts an inhibitory effect on inflammatory signaling pathways by modulating the NF-κB signaling pathway. Dysfunction of KRT32 leads to an imbalance in keratinocyte immunoregulation, contributing to the development of PRP. Our results have provided both genetic and immunological evidence, underscoring the crucial role of keratinocyte-specific gene KRT32, in the regulation of skin immune homeostasis. This finding offers valuable insights into the underlying etiology and pathomechanisms of PRP.

## Results
### WES identified damaging variants of *KRT32* in patients with PRP
In the discovery stage, whole exome sequencing (WES) was performed on 58 patients with PRP and 364 healthy controls to investigate the genetic basis underlying the PRP. We totally obtained 15,988 rare damaging variants, including 8397 genes with a minor allele frequency (MAF) < 0.0005 in the ExAC-ASN databases after applying a series of filtering strategies. A heterozygous mutation

c.1799C>T (p.Pro600Leu), in the *CARD14* gene was identified. To identify potentially pathogenic genes associated with PRP, a gene burden test was conducted on all 8397 genes after excluding one case with damaging *CARD14* variants. Our primary focus was on genes specifically expressed in keratinocytes among the top genes associated with PRP (Supplementary Data 1). Intriguingly, the *KRT32* gene, which encodes keratin, exhibited the most significant association with PRP ($P = 3.06 \times 10^{-4}$), reinforcing the critical role of keratinocytes in PRP pathogenesis. Among the 57 cases analyzed, four individuals (7.02%) carried *KRT32* mutations (Table 1; individual 1: c.344G>A (p.Arg115Gln), individual 2: c.477_478del (p.Thr160fs), individual 3: c.607C>T (p.Arg203Cys), and individual 4: c.685T>C (p.Cys229Arg)). None of the mutations were detected in the 364 healthy controls.

To validate this finding, an additional cohort consisting of 44 PRP patients and 436 healthy controls was included in the validation analysis using Sanger sequencing. The analysis identified two damaging variants in the *KRT32* gene (Table 1; individual 5: c.907G>A (p.Glu303Lys), individual 6: c.937A>G (p.Ile313Val)) that exhibited significant differences when compared to controls ($P = 8.23 \times 10^{-3}$). A combined analysis of the *KRT32* gene in both the discovery and validation cohorts revealed a significant p value of $1.73 \times 10^{-6}$ (Table 2). Supplementary Table 1 and Fig. 1 outline the clinical features of PRP patients with damaging mutations. All the 102 PRP patients' clinical manifestations have been incorporated into Supplementary Data 2 and Supplementary Table 2.

### The structural basis of functional abnormality caused by KRT32 mutations
Six damaging variants of *KRT32* in patients with PRP were further verified by Sanger sequencing (Fig. 2A). The KRT32 protein consists of a central alpha-helical rod domain flanked by non-alpha-helical N-terminal (head) and C-terminal (tail) domains with the helical rod domain being divided into Coil 1A, 1B, and 2 subdomains by linker sequences to form the intermediate filament (IF) rod domain. We subsequently charted the distribution of the six variants across different sub-domains (Fig. 2B) and annotated them on a 3D structure simulation of KRT32 protein (Fig. 2C). Our finding revealed that all variants were concentrated in Coil subdomains (Coil 1A, Coil 1B, and Coil 2), which are known to frequently facilitate protein-protein interactions[24]. To evaluate the potential pathogenic impact of the six mutations, a homologous protein alignment program was employed to compare KRT32 protein sequences across various species. As shown in Fig. 2D, all the mutations were exclusively located within conserved regions, implying that these six variants may significantly impair the functionality of KRT32 protein.

### KRT32 damaging mutations promote cell proliferation and alter their expression patterns in the skin layers of PRP patients
To investigate the tissue-specific expression pattern of KRT32, we analyzed publicly available datasets, including HPA (Human Protein

**Table 1 | Genetic finding in PRP patients**

| Gene.refGene | AAChange | ExonicFunc.refGene | SIFT score | Polyphen2 HDIV_score | Polyphen2 HVAR_score | MutationTaster score | ExAC_EAS |
|---|---|---|---|---|---|---|---|
| KRT32 | c.344G>A(p.Arg115Gln) | missense | D | D | D | D | . |
| | c.477_478del(p.Thr160fs) | frameshift substitution | . | . | . | . | . |
| | c.607C>T(p.Arg203Cys) | missense | D | D | D | D | . |
| | c.685T>C(p.Cys229Arg) | missense | D | D | D | D | $5.00 \times 10^{-4}$ |
| | c.907G>A(p.Glu303Lys) | missense | D | D | D | D | . |
| | c.937A>G(p.Ile313Val) | missense | D | P | D | D | . |
| CARD14 | c.1799C>T(p.Pro600Leu) | missense | D | D | D | D | . |

*D* deleterious, *P* possible damaging, *N* neutral.

Atlas) and GTEx (Genotype-Tissue Expression). As shown in Supplementary Fig. 1A–C, KRT32 exhibited high expression in the tissues of skin and esophagus, especially in the basal keratinocytes of human skin. Additionally, our immunohistochemical (IHC) analysis revealed distinct expression patterns of mutant KRT32 of PRP patients

### Table 2 | Carrying rate of the damaging mutations of *KRT32* in Case participants versus Control Subjects

| | Genotype Counts (%) | | |
| --- | --- | --- | --- |
| | Cases | Controls | P Value |
| Discovery | 4/57 (7.02%)[a] | 0/364 (0%) | $3.06 \times 10^{-4}$ |
| Validation | 2/44 (4.55%) | 0/436 (0%) | $8.23 \times 10^{-3}$ |
| Combination | 6/101 (5.94%) | 0/800 (0%) | $1.73 \times 10^{-6}$ |

[a]The case carrying the pathogenic variant of *CARD14* has been rejected. *P*-values were calculated using two-sided Fisher's exact test.

compared to that in healthy controls and PRP patients without *KRT32* mutations (Supplementary Fig. 1D). In health controls, KRT32 was predominantly expressed in the basal keratinocytes. However, in PRP patients with *KRT32* mutations, the epidermal layer was thickened, and KRT32 showed a uniform expression in both basal and supra-basal keratinocytes. Interestingly, although PRP patients without *KRT32* mutations showed a thicken epidermal layer, KRT32 was still prominently expressed in basal keratinocytes rather than supra-basal keratinocytes. Thus, the damaging KRT32 mutations changed the expression location itself in the skin of PRP patients.

The thickened epidermal layer of skin lesion indicated the abnormal proliferation of keratinocyte cells, and we found a much higher ratio of Ki67-positive cells in PRP patients with *KRT32* mutation compared to healthy controls (Fig. 3A). To validate the role of KRT32 in cellular proliferation, immortalized keratinocytes cell lines (Ker-CT), were used to build the cell models with overexpression and knockdown of KRT32, as depicted in Supplementary Fig. 2. Cell Counting Kit-

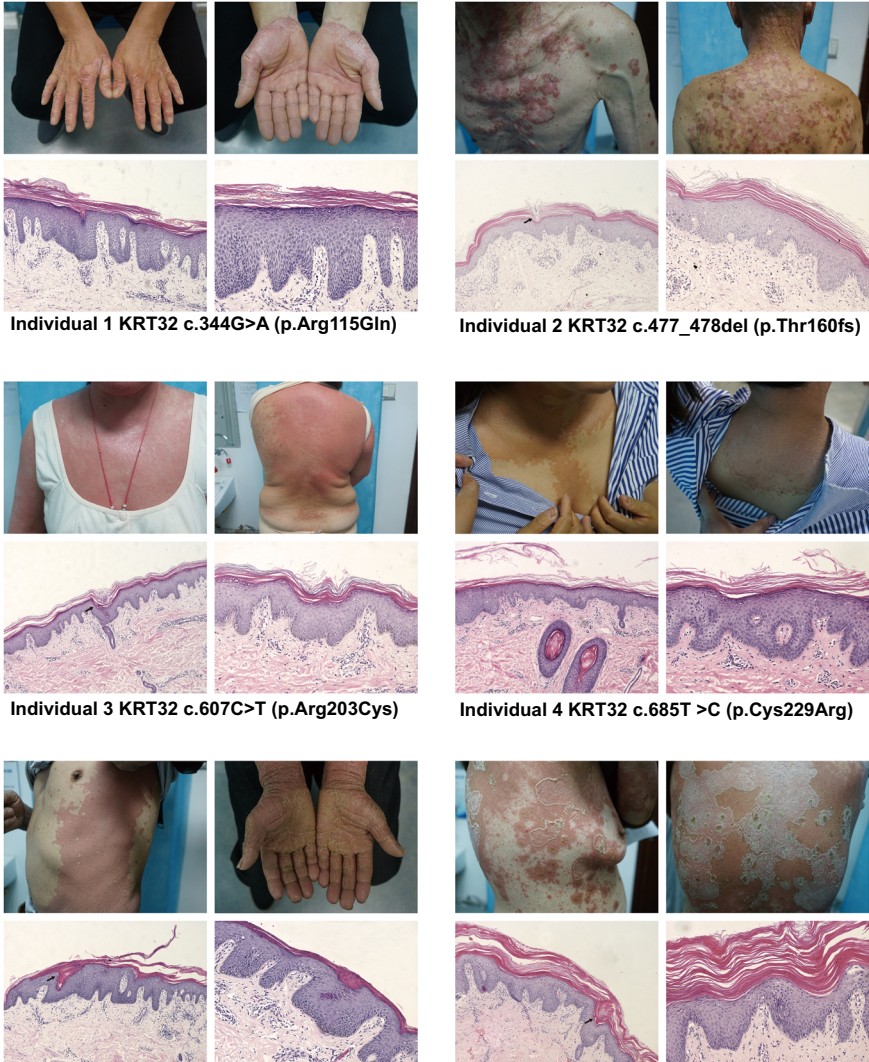

**Individual 1 KRT32 c.344G>A (p.Arg115Gln)**

**Individual 2 KRT32 c.477_478del (p.Thr160fs)**

**Individual 3 KRT32 c.607C>T (p.Arg203Cys)**

**Individual 4 KRT32 c.685T >C (p.Cys229Arg)**

**Individual 5 KRT32 c.907G>A(p.Glu303Lys)**

**Individual 6 KRT32 c.937A>G (p.Ile313Val)**

**Fig. 1 | Clinical and histopathological features of six PRP patients with *KRT32* mutations.** Photographs of cutaneous lesions and histological analysis of skin biopsy specimens from PRP patients with *KRT32* mutations are presented. These patients manifested typical pathological characteristics of PRP, including widespread erythematous plaque coalescence, layered parakeratosis, absence of Munro microabscesses, alternating hyperkeratosis and parakeratosis, irregular thickening of the stratum spinosum, sparse lymphoid cell infiltration around superficial dermal blood vessels, and hair follicular plugging (labeled by black arrows). Each group of H&E-stained histological images are captured at two magnification levels – 100× (left) and 200× (right), and the mutation site is presented below the respective images. The process of producing these representative H&E-stained histological images was repeated, yielding similar results for all the 3 slides (*n* = 3) per patient.

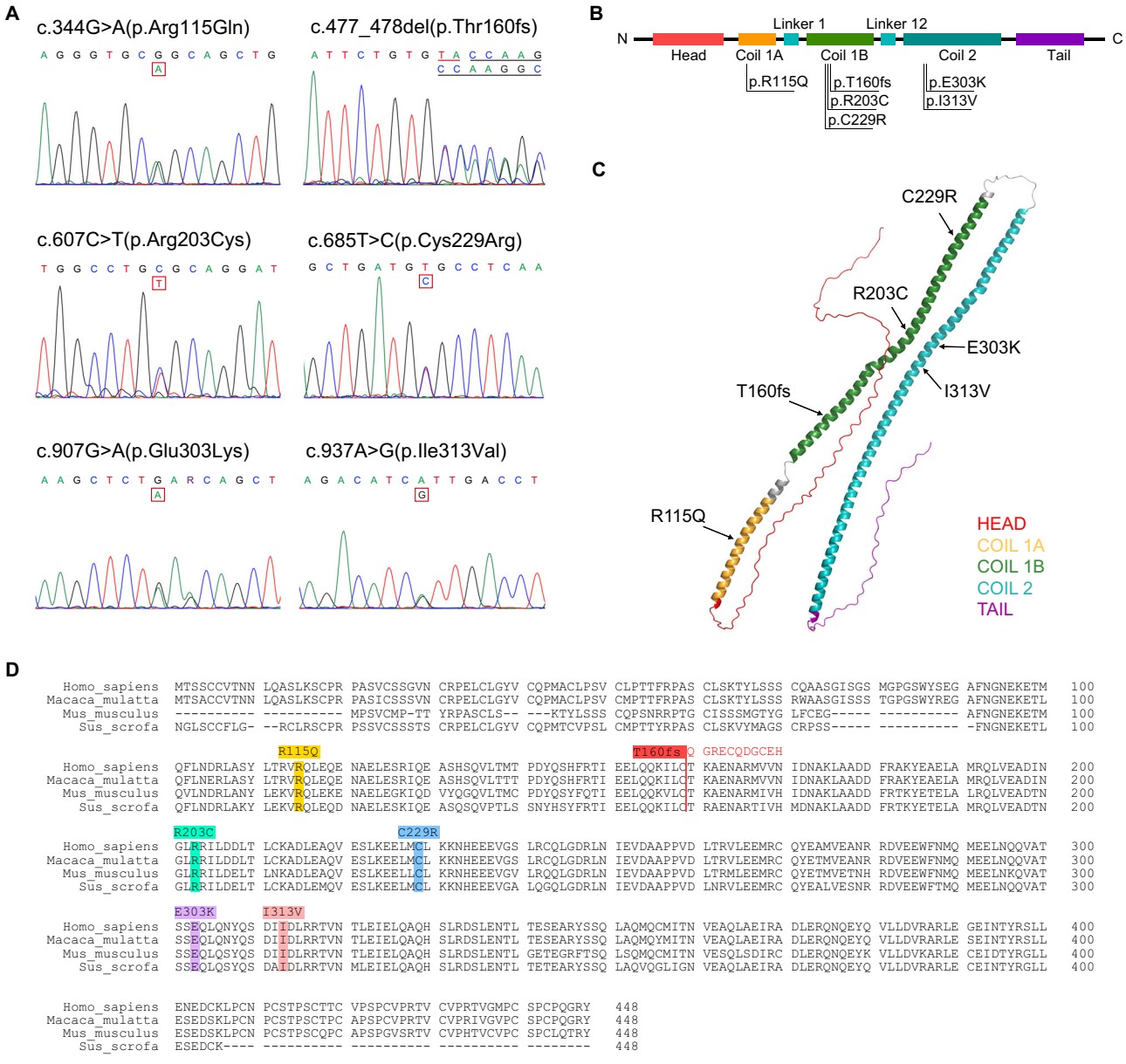

**Fig. 2 | The locations of damaging mutation sites of KRT32 protein from PRP patients. A** Six mutations of *KRT32* in patients with PRP are shown by Sanger sequencing traces. **B** The schematic primary structure of KRT32 with the six pathogenic mutations. **C** Mapping the mutations of KRT32 on the 3D structure predicted by the AlphaFold software (available at https://alphafold.ebi.ac.uk/).

**D** Sequence alignment of Homo sapiens KRT32 proteins with those of Macaca mulatta (XP_014975129.2), Mus musculus (NP_001152846.2), and Sus scrofa (XP_003131481.2), respectively. Six pathogenic mutation sites are marked with different colors.

8 (CCK-8) assays demonstrated that knockdown of KRT32 facilitates keratinocyte proliferation (Fig. 3B), whereas overexpression of KRT32 wildtype significantly suppressed keratinocyte cell proliferation (Fig. 3C). However, all damaging KRT32 mutations lose the ability to suppress cell proliferation compared to KRT32 WT in keratinocyte cells (Fig. 3C). Furthermore, 5-ethynyl-2′-deoxyuridine (EdU) assays also validated this conclusion (Fig. 3D, E). In summary, all the findings suggest that KRT32 plays a pivotal role in the suppression of the keratinocyte cell proliferation, but KRT32 damaging mutations in PRP lose such function.

## KRT32 is a negative regulator of NF-κB-mediated inflammatory response

To investigate the mechanism of how KRT32 damaging variants affect the progression of PRP disease, we performed RNA sequencing using

the formalin-fixed paraffin-embedded (FFPE) samples obtained from six PRP patients with *KRT32* mutations and six healthy controls with patient-matched normal skin tissues. The KEGG enrichment assay identified significant enrichment of NF-κB and TNF signaling pathways (Supplementary Fig. 3A and Supplementary Data 3), with differentially expressed genes showing a significant upregulation of pro-inflammatory gene including TNF, IL1B, ILA and IL8 (CXCL8) (Supplementary Fig. 3B and Supplementary Data 4). These results align with previous investigations implicating the NF-κB pathway in the pathogenesis of PRP[6]. Subsequently, we assessed the levels of NF-κB-related inflammatory factors (TNF, IL-6, IL-1β, and IL-8) in the blood and skin of PRP patients. Interestingly, only serum TNF levels in PRP patients showed a significant elevation compared to those in healthy controls (Supplementary Fig. 3C). Furthermore, healed PRP patients exhibit much lower serum TNF levels compared to levels before treatment

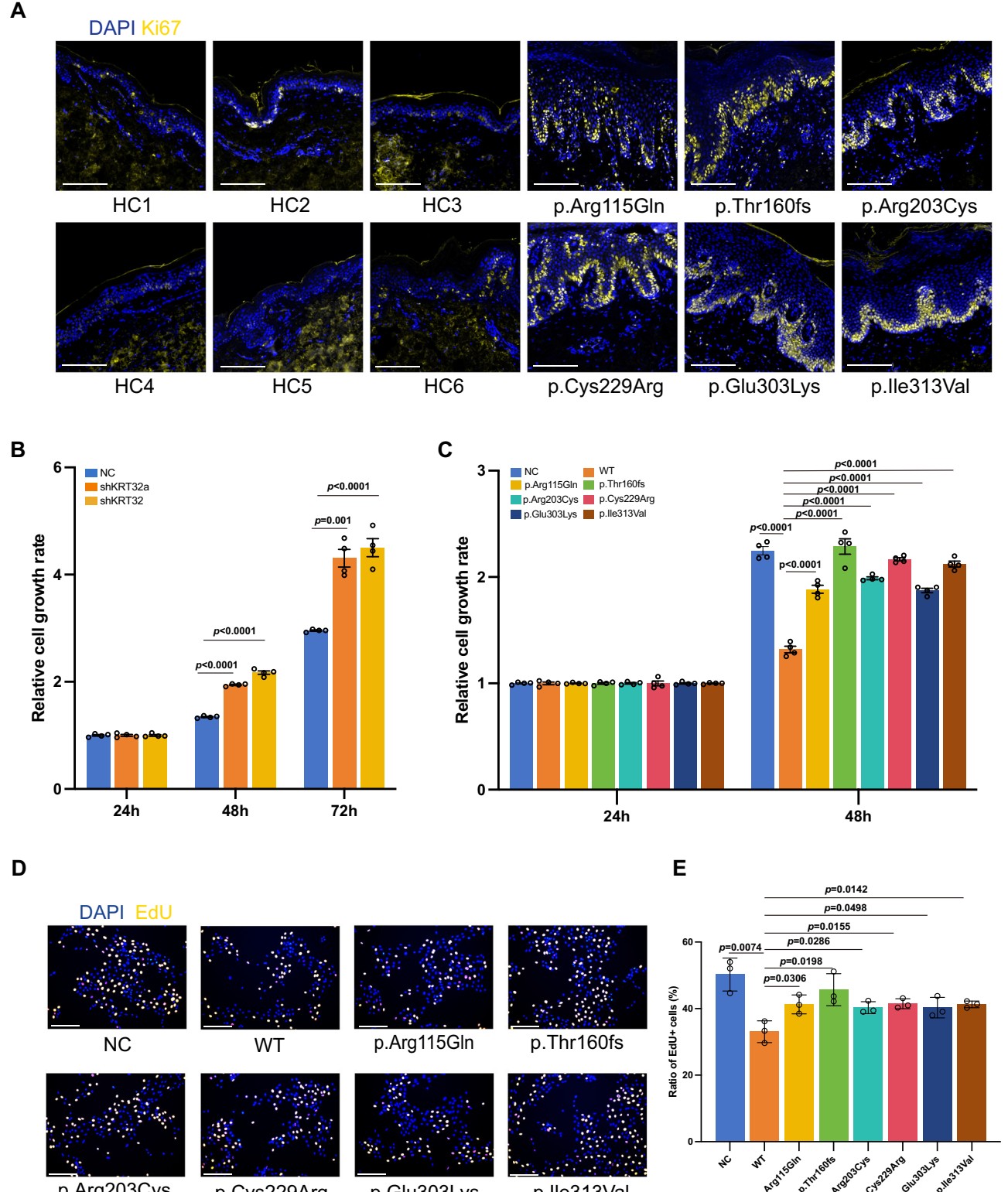

**Fig. 3 | Loss-of-function KRT32 mutations promote cell proliferation.**
**A** Representative immunofluorescence staining for Ki67 of skin biopsy sections obtained from patients with *KRT32* mutation and matched healthy controls (HC; *n* = 6). Scale bar represent 150 µm. Age, gender, and skin sampling site matching were ensured between healthy individuals and patients. CCK-8 assays to assess the function of KRT32 on cell proliferation through KRT32 knockdown (**B**) or wildtype (WT) and six KRT32 mutations overexpression (**C**) in Ker-CT cells. shKRT32a and shKRT32 represent two different shRNA sequences against KRT32. All sample OD450 values were normalized to their relative samples at 24 h. Data are shown as means ± SEM of four independent biological repeats. *P* value was calculated using ordinary one-way ANOVA with Dunnett's multiple comparisons test in (**B**) and (**C**). EdU assay to evaluate the influence of KRT32 mutations on Ker-CT cell proliferation. One representative experiment shown in (**D**). Scale bars represent 75 µm. Statistical analysis of EdU-positive cells in three independent biological repeats (**E**). Data were shown as means ± SEM, and *P* value was calculated using a two-sided unpaired Student's *t* test. Source data are provided as Source Data file.

(Supplementary Fig. 3D). Additionally, the serum TNF level of PRP patients show positively correlated with the disease severity (Pearson correlation 0.7273, $P < 0.0001$), as indicated by the established individual PRP area and severity index[25] (Supplementary Fig. 3E). These results suggest the potential role of TNF as a trigger for PRP. Immunohistochemistry (IHC) assays further revealed a significant upregulation of pro-inflammatory cytokines (TNF, IL-1β, IL-6, and IL-8) activation in the epidermis of PRP lesions (Supplementary Fig. 4). The inflammatory response observed in the skin was more pronounced compared to that of the circulatory system, indicating the potentially pivotal role of epidermal keratinocytes immune homeostasis in patients with the pathogenesis of PRP.

Subsequently, transcriptomic analysis was performed on Ker-CT cells overexpressing KRT32. The results revealed that KRT32 overexpression suppressed TNF and NF-κB signaling pathways (Fig. 4A and Supplementary Data 5), as well as the expression of numerous proinflammatory gene (such as TNF, IL1A, IL1B, IL8, IL6, etc.) (Fig. 4B and Supplementary Data 6). Additionally, KRT32 overexpression significantly inhibited NF-κB activation in a dose-dependent manner compared to mock control via luciferase reporter activity assay (Fig. 4C). KRT32 knockdown increased p65 phosphorylation levels (Fig. 4D) and IL-1β, IL-6, IL-8 secretion in Ker-CT cells (Fig. 4E–G), on the contrary KRT32 overexpression reduced p65 phosphorylation levels in primary keratinocytes cells and Ker-CT cells (Fig. 4H, I) and IL-1β, IL-6, IL-8 secretion in Ker-CT cells (Fig. 4J–L). Compared to wild-type KRT32, the harmful mutations promoted p65 phosphorylation (Fig. 4M) as well as IL-1β and IL-8 secretion (Fig. 4N, O). Subsequently, we evaluate the impact of the *KRT32* damaging variants, as well as benign and 12 common (MAF > 0.0005) variants, on the NF-κB activation relative to wild-type KRT32 using the NF-κB luciferase reporter system. Rare (MAF < 0.0005) and damaging *KRT32* variants significantly upregulated TNF-induced NF-κB activation compared with wild-type protein. In contrast, all the benign and common variants did not exhibit a significant impact on the NF-κB signaling pathway (Fig. 4P). In summary, our findings confirmed that KRT32 is a negative regulator of NF-κB in the epidermis, and that loss-of-function mutations resulting from damaging variants lead to hyperactivation of NF-κB.

## KRT32 binds directly to NEMO

To investigate the molecular mechanisms underlying KRT32-related negative regulation of the NF-κB signaling pathway, we employed GST-KRT32 pulldown assays coupled with mass spectrometry to identify KRT32-associated proteins from lysates of primary keratinocytes cells (Fig. 5A). Among the proteins identified in the GST-KRT32 immunoprecipitated complex through mass spectrometry, the NF-κB essential modulator NEMO/IKKγ, encoded by the IKBKG gene, was identified (Supplementary Data 7). Further validation of the interaction between KRT32 and NEMO was achieved in both primary keratinocyte cells and Ker-CT cell line by western blot (Fis. 5B, C). In HEK293T cells co-expressing HA-KRT32 and Flag-tagged NEMO or other NF-κB components, co-immunoprecipitation was performed to confirm the interactions between KRT32 and NEMO (Supplementary Fig. 5A). The GST pull-down assay showed that GST-KRT32 binds directly to NEMO, rather than IKKα, IKKβ, and IκBα (Supplementary Fig. 5B). Co-IP experiments further confirmed the interaction between endogenous NEMO and KRT32 in both of primary keratinocytes cells and Ker-CT cells (Fig. 5D, E). Additionally, proximity ligation assays (PLA) were utilized to further confirm the physical proximity of endogenous KRT32 and NEMO in human skin (Fig. 5F) as well as in primary keratinocytes cells and Ker-CT cells (Fig. 5G, H). Next, we utilized a well-recognized NEMO inhibitor, NBD (IKKγ/NEMO-binding domain), to reduce the function of NEMO in NK-κB by disrupting its association with IKKβ[26,27]. When the binding of NEMO and IKKβ is inhibited by NBD, KRT32 mutations restore the level of IL-1β secretion to the same level of KRT32 overexpression cells. And NBD treatment of the NC cells

shows a similar IL-1β secretion to the KRT32 overexpressed cells in the context of NBD treatment (Fig. 5I). Together, all the data demonstrated that KRT32 is a key factor inhibiting the activation of NF-κB pathway through the regulation of NEMO protein.

## Damaging mutations in KRT32 impair its binding ability to NEMO

To determine the binding mode, the docking HDCOK program was used to visualize the interactions interface region between KRT32 and NEMO proteins. The optimal modeling results are shown in Fig. 6A (docking score = −326.04, confidence score = 0.896). The C-terminus of the NEMO protein binds to the gap between the two long helices of the KRT32 protein molecule, resulting in a robust interaction between these two molecules. Further analysis revealed that Arg115, Arg203 and Glu303 in the six identified mutation sites on the KRT32 protein directly formed intermolecular hydrogen-bond interactions with NEMO. Although the other three mutation sites (lle313, Thr160, and Cys229) do not directly conjugate to the binding sites of NEMO, they are located within the interacting interface. Therefore, we supposed that mutations at these sites may lead to the changes in domain conformation and then weaken the interaction between KRT32 and NEMO (Fig. 6B). Furthermore, an in vitro protein binding experiment further confirmed that the diminished interacting ability of KRT32 mutations to bind to NEMO was impaired to a certain extent (Fig. 6C). And PLA analysis additionally validated such results in Ker-CT cells (Fig. 6D, E).

## KRT32 promotes the K48-linked polyubiquitination-mediated degradation of NEMO protein and impedes the assembly of IKK complex

mIHC assay shows high NEMO expression in the epidermis of PRP lesions in patients with *KRT32* mutations compared with healthy individuals and PRP without *KRT32* mutations (Supplementary Fig. 6), leading us to hypothesize that KRT32 may regulate NEMO expression. Although KRT32 overexpression or knockdown did not change mRNA level of IKBKG/NEMO in Ker-CT cells (Fig. 7A, B), it changed its protein expression (Fig. 7C, D). Furthermore, damaging mutations in KRT32 elevated the protein level of NEMO compared to the wild type (Fig. 7E), suggesting a potential role of KRT32 in regulating NEMO protein stability. Subsequently, we inhibited de novo NEMO synthesis using protein synthesis inhibitor cycloheximide (CHX) for 6 and 12 h, and found that KRT32 overexpression accelerated the degradation of NEMO protein (Fig. 7F). Subsequently, a proteasome inhibitor MG132 indeed stabilized endogenous NEMO protein in Ker-CT cells overexpressing KRT32 (Fig. 7G). Therefore, we supposed that KRT32 expression might affect the stability of NEMO via ubiquitin proteasome system. We further confirmed that KRT32 protein facilitates the ubiquitination of NEMO, as confirmed by Ub antibody immunoblotting analysis (Fig. 7H). Furthermore, ubiquitination assays revealed that polyubiquitination of NEMO was mediated by the formation of a chain linked via lysine 48 in NEMO (Fig. 7I). In comparison to the wild-type, KRT32 mutations diminished K48 polyubiquitination of NEMO (Fig. 7J). Consistently, PLA also indicates that damaging variants of KRT32 overexpression in Ker-CT cells impaired the co-localization of NEMO with Ub molecules compared to the wild-type (Fig. 7K, L).

NEMO, along with IKKα and IKKβ subunits, forms the IKK complex, which is an essential complex for NF-κB activation[28]. Thus, we evaluated the effect of KRT32 WT and its mutants on the assembly of the IKK complex. PLA experiment suggested that wild-type KRT32 overexpression significantly reduced the interaction between IKKα and NEMO, whereas harmful mutations in KRT32 strengthen the combination of IKKα and NEMO compared to the wild-type (Fig. 7M, N). Moreover, the in vitro pull-down assay demonstrated that KRT32 impedes the assembly of NEMO complexes with IKKα/IKKβ, while damaging variants markedly attenuated the inhibitory effect on the formation of the IKK complex (Fig. 7O, P). Consistently, KRT32 mutations led to a higher level

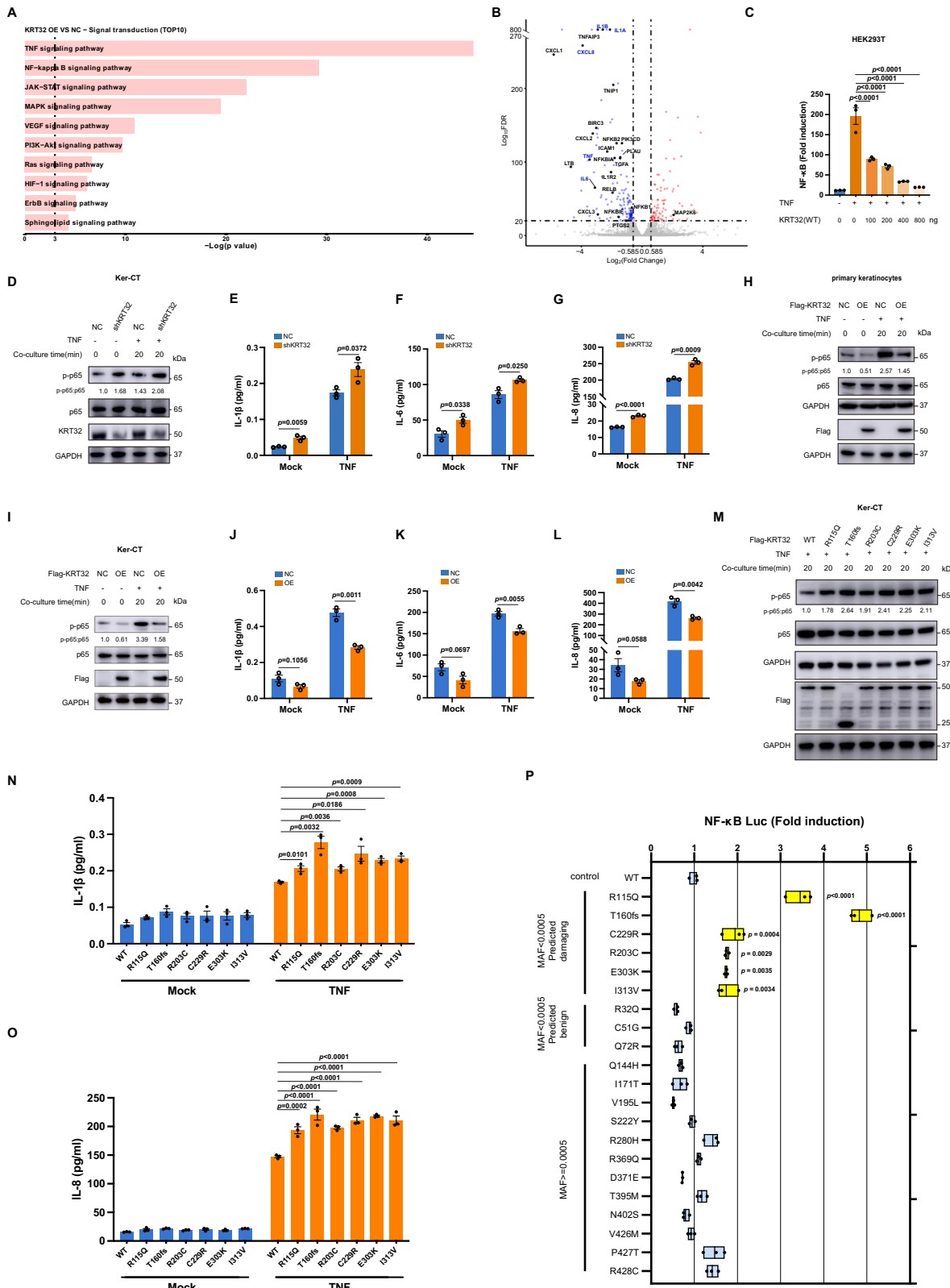

of phosphorylation of IKKα/IKKβ compared to the overexpression of KRT32 wild-type protein in Ker-CT cells (Fig. 7Q). All the above findings suggested that KRT32 inhibits IKKα/IKKβ phosphorylation and IKK complex formation by binding to NEMO and subsequent UPS-mediated protein degradation, whereas harmful mutations lose such functions, leading to the overactivation of NF-κB.

## *Krt32* knockout in mice recapitulates the PRP-like dermatitis phenotype

To further investigate the effects of the loss-of-function mutations of *KRT32* from PRP patients, we generated a *Krt32* knockout mouse model (C57BL/6J) using the CRISPR-Cas9 approach. And the *Krt32* knockout mice were confirmed by Sanger sequencing (Supplementary Fig. 7).

**Fig. 4 | KRT32 inhibits the activation of NF-κB signaling pathway. A** Selected KEGG signal transduction pathway enrichment analysis with significant DEGs (FDR < 0.05 and log$_2$FC < −0.585) through RNA sequencing of Ker-CT cells with KRT32 overexpression. The statistical test was hypergeometric test, and the level of significance was set at a two-sided $P$ < 0.05 without multiple comparisons. **B** Volcano plot illustrating significant DEGs between KRT32 overexpression and negative control in Ker-CT cells. Upregulated genes were labelled by red dots with FDR < 1e$^{-20}$, log$_2$FC > 0.585; and downregulated genes was labelled by blue dots with FDR < 1e$^{-20}$, log$_2$FC < −0.585. Selected NF-κB pathway-associated genes are highlighted. FC denotes fold change. The statistical test was Wald test and the level of significance was set at a two-sided FDR < 0.05 with multiple comparisons by Benjamini-Hochberg method. **C** NF-κB-dependent luciferase activation assay in HEK293T cells to analyze concentration-dependent role of KRT32 in NF-κB activation under the stimulation with TNF (20 ng/mL). **D** The phosphorylated p65 expression in KRT32 knockdown Ker-CT cells with and without TNF (20 ng/mL) treatment for the indicated time. **E–G** Detection of IL1β, IL-6, IL8 secretion in supernatant of KRT32 knockdown Ker-CT cells with or without TNF treatment by MSD assay. **H–I** The phosphorylated p65 expression in primary keratinocytes and

Ker-CT cells with Flag-tagged KRT32 overexpression. **J–L** MSD assay detection of IL1β, IL-6, IL8 secretion of Ker-CT cells overexpressing KRT32. **M** The phosphorylated p65 expression in Ker-CT cells overexpressing Flag-tagged KRT32 wildtype and mutations. **N, O** MSD assay detection of IL1β, IL-8 secretion of Ker-CT cells overexpressing KRT32 wildtype and mutations with and without TNF treatment. **P** Analysis of luciferase activity after transfection with expression vectors encoding KRT32 wildtype and the variants (6 rare (MAF < 0.0005) and predicted to be damaging variants, 3 rare (MAF < 0.0005) and predicted to be benign variants, and 12 common (MAF > 0.0005) variants) in stimulated with TNF (20 ng/mL) for 12 h. Data are shown as means ± SEM of three independent biological repeats in (**C**, **E–G**, **J–L** and **N–P**). The $P$ value was calculated using ordinary one-way ANOVA with Dunnett's multiple comparisons test in (**C**, **P**); a two-sided unpaired Student's $t$ test in (**E–G**, **J–L**); and two-ANOVA with Dunnett's multiple comparisons test in (**N**) and (**O**). Floating bars show the minimum, average, and maximum values within each group in (**P**). One representative experiment from two independent experiments with similar results is shown in (**D**, **H**, **I** and **M**). Source data are provided as Source Data file.

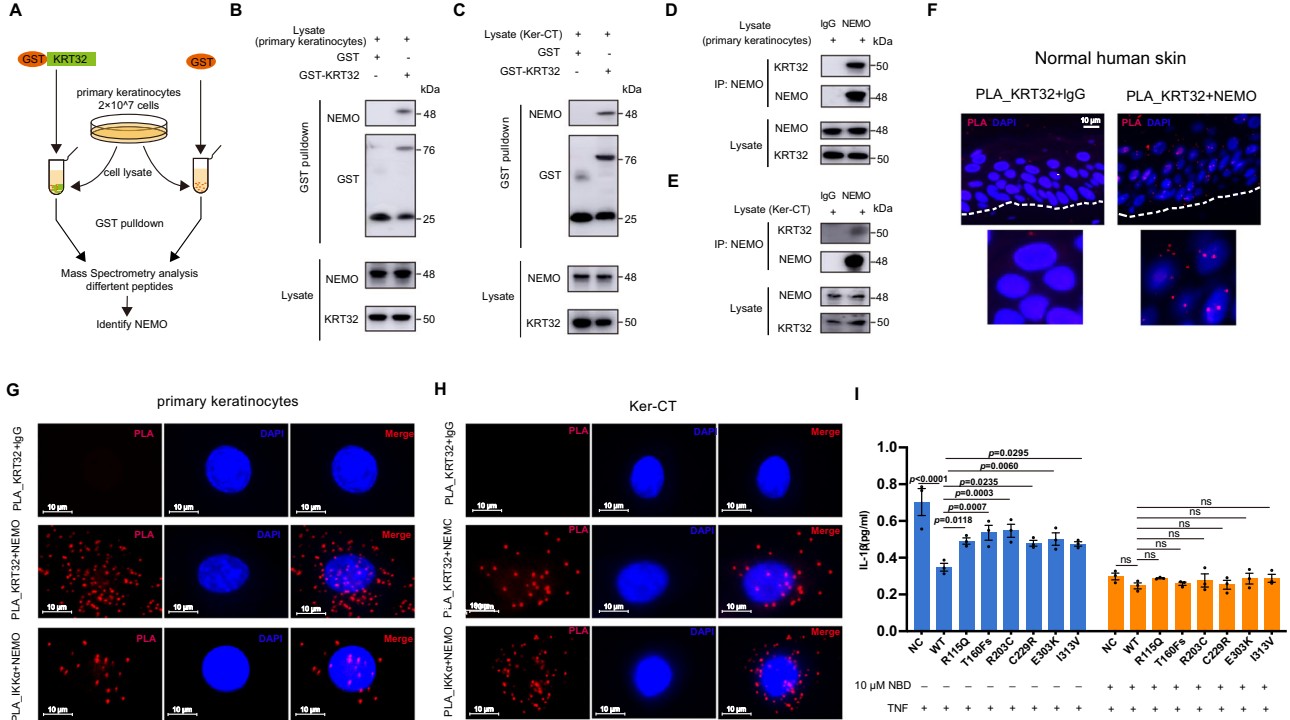

**Fig. 5 | KRT32 interacts with NEMO. A** Schematic diagram of the procedure to identify KRT32 interaction proteins in primary keratinocytes cells. *Escherichia coli* (*E. coli*) extracts containing GST (negative control) or GST-KRT32 proteins were incubated with cell lysates of primary keratinocytes cells and glutathione-Sepharose beads to pull down GST complexes, and the bound proteins were then analyzed by mass spectrometry. **B, C** GST and GST-KRT32 proteins were incubated with lysates from primary keratinocytes cells and Ker-CT cells followed by Western blot analysis using anti-NEMO antibodies. **D, E** Co-IP analysis of the endogenous interaction of NEMO using anti-NEMO antibody in primary keratinocytes cells and Ker-CT cells. **F** PLA analysis of endogenous KRT32 and NEMO interaction in normal

human adult skin. Scale bars represent 10 µm. **G, H** PLA of endogenous KRT32 and NEMO in primary keratinocytes cells and Ker-CT cells. IgG was used as negative controls, and IKKα-NEMO interaction was used as positive control. Scale bars represent 10 µm. **I** MSD assay detection of IL-1β secretion of Ker-CT cells over-expressing KRT32 wildtype and mutations with and without 10 µM NBD-inhibitory peptide for 2 h. Data shown as means ± SEM of three independent biological repeats in (**I**). $P$ value was calculated using a two-way ANOVA with Dunnett's multiple comparisons test (ns; no significant difference). One representative experiment from two independent experiments with similar results is shown in (**B–H**). Source data are provided as Source Data file.

The process of the mouse modeling and experimental workflow for PRP-like mice were present in Fig. 8A.

Initially, the epidermal appearance of *Krt32*$^{(-/-)}$ mice and their offspring did not show obvious alterations compared to that of WT mice under normal conditions. However, upon TNF treatment for 2 days, the dorsal skin of *Krt32*$^{(-/-)}$ mice exhibited obvious thickness and extensive yellow scaling, resembling the cutaneous manifestations observed in human patients with PRP (Fig. 8B and

Supplementary Fig. 8A). Histological analysis of dorsal skin lesion sections from *Krt32*$^{(-/-)}$ mice exhibited a thickened epidermis, hyperkeratosis with focal parakeratosis, and mild lymphocyte infiltration. Notably, a hair follicle plugging-like change was also observed in KO mice, resembling a typical feature of PRP (Fig. 8C and Supplementary Fig. 8B). We further analyzed the microscopic structures of the hair and nail using scanning electron microscopy (SEM). The surface of hair of *Krt32* KO did not show significant

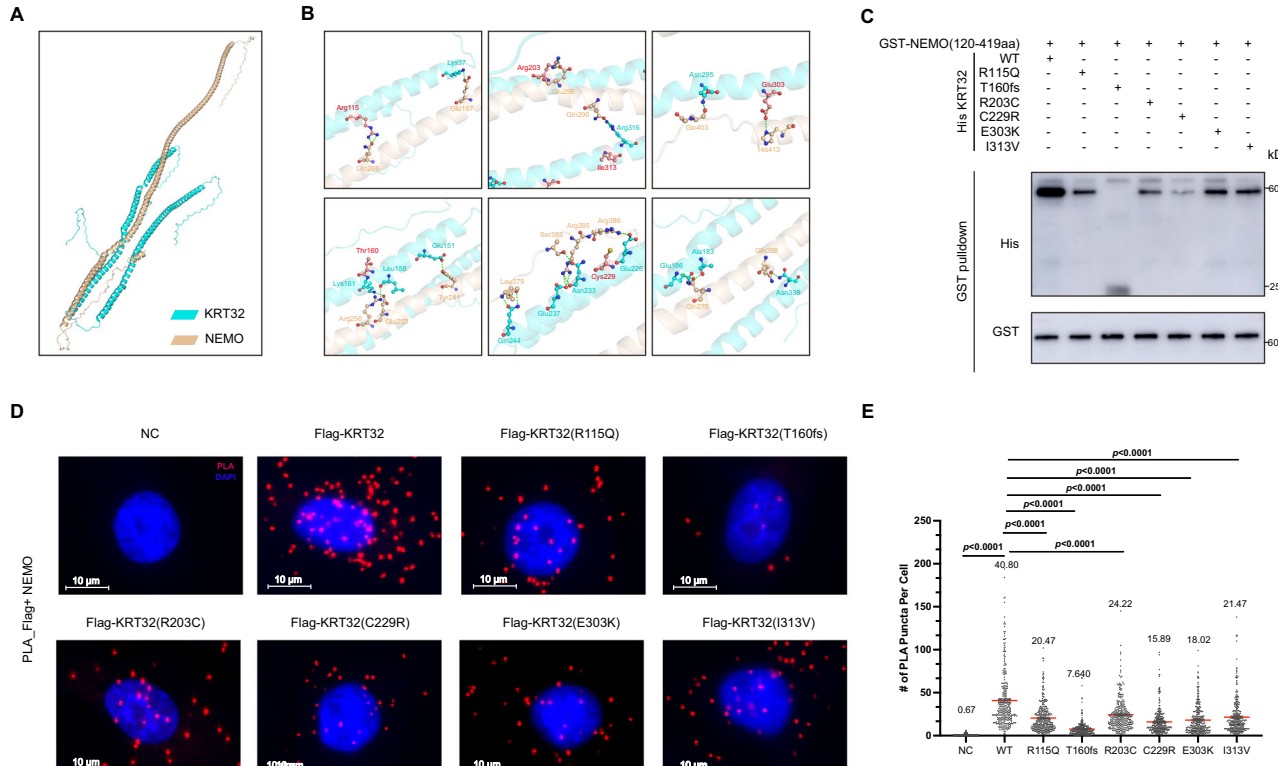

**Fig. 6 | The binding of NEMO with damaging KRT32 mutations. A** The molecular docking software HDOCK was used to simulate the interaction between KRT32 and NEMO. **B** Detailed interactions of KRT32 with NEMO. The confirmed active sites of KRT32 are shown as red sticks, while the other residues critical for binding are shown as blue sticks (KRT32) and golden sticks (NEMO). The hydrogen bond is shown as a green dashed line. **C** GST-NEMO fusion protein was incubated with *E. coli* extracts containing WT or six mutant His-KRT32 proteins. GST complexes were isolated with glutathione-agarose beads, and the interacting proteins present in the pull-down eluate were detected by western blotting. **D** PLA analysis of the interaction of Flag-tagged KRT32 wildtype and mutations with NEMO in Ker-CT cells. Scale bars represent 10 μm. **E** Quantification of the number of PLA foci per cell in (**D**). Each dot on the graph corresponds to a specific analyzed cell. Red bars represent the mean ± SEM from the indicated number of cells (*N*). The number of cells analyzed per group varies as follows: *N* = 311, 299, 335, 292, 312, 303, 303, and 311, with each group consisting of three biological replicates. *P* value was calculated using a two-sided unpaired Student's *t* test. Source data are provided as Source Data file.

changes, but the surface of nail becomes very rough (Supplementary Fig. 9A, B). Thus, *Krt32* KO recapitulates the PRP-like phenotypes in humans, which supported our idea that KRT32 plays a pivotal role in the development of PRP disease.

To further confirm whether the TNF-treated *Krt32*$^{(-/-)}$ mice could recapitulate the gene expression patterns of human PRP diseases, we conducted the RNA-seq analysis of the dorsal skin from *Krt32* KO and WT mice. KEGG analysis revealed the enriched NF-κB and TNF pathways in *Krt32* knockout mice (Fig. 8D, Supplementary Data 8), consisting with the results from PRP patients with *KRT32* mutations. The gene transcription of pro-inflammatory cytokines TNF, IL-1β, and IL-6 showed significant upregulation in KO mouse (Fig. 8E, Supplementary Data 9), RT-qPCR also validated these results (Fig. 8F, G). Meanwhile, IHC analyses demonstrated that IL-1β expression levels were elevated in the epidermis of PRP mouse model (Fig. 8H and Supplementary Fig. 8C).

We have demonstrated that KRT32 inhibited the unscheduled keratinocyte cell proliferation and regulated the suppression of NF-κB via the interaction with NEMO in skin tissue and keratinocyte cells. Consistently, the PRP mouse model also manifested thicker epidermis with high expression of Ki67 and NEMO, confirmed by IHC (Fig. 8I, J and Supplementary Fig. 8D, E). Furthermore, immunoblotting showed elevated levels of NEMO and phosphorylated p65 in both mouse skin tissue lysates and sucking mouse epidermal cell lysates due to *Krt32* deficiency (Fig. 8K–M). Thus, KRT32 in mouse epidermal cell interacts with NEMO to inhibit the activation of NF-κB pathway and maintain skin immune homeostasis.

## Discussion

KRT32 belongs to the family of keratins, a type of intermediate filament (IF) proteins that form intricate filament systems in epithelial cells, thereby providing structural support to keratinocytes for maintaining skin integrity[29,30]. However, electron microscopy analysis conducted in the present study demonstrated that loss of KRT32 exerts no discernible impact on the structural integrity of keratinocytes (desmosomes and hemidesmosomes) in the PRP-like dermatitis *Krt32*$^{(-/-)}$ mice model (Supplementary Fig. 10). Here, we provide evidence that KRT32 plays a unique role in maintaining immune homeostasis of skin by functioning as a negative regulator inflammatory response within the NF-κB pathways (Supplementary Fig. 11). This study enhances our understanding of the function of KRT32 and sheds light on the pathogenic mechanisms underlying PRP.

PRP is a rare chronic inflammatory papulosquamous dermatosis characterized by epidermal hyperkeratosis. In our study, we found KRT32 regulates keratinocyte proliferation, as evidenced by the thickened epidermis, and hyperkeratosis with focal parakeratosis observed in *Krt32*$^{(-/-)}$ mice. These findings are consistent with the expression pattern of KRT32, which is primarily localized to basal keratinocytes in the skin and minimally expressed in other immune cells. Basal keratinocytes serve as mitotically active progenitor cells, gradually differentiating into cells of the upper layers. We hypothesize that KRT32 in the basal layer regulates skin cell proliferation and differentiation, with loss-of-function mutations of KRT32 in PRP patients inhibiting basal cell differentiation and maintaining strong proliferation in the spinous layer. Consequently, PRP patients present with

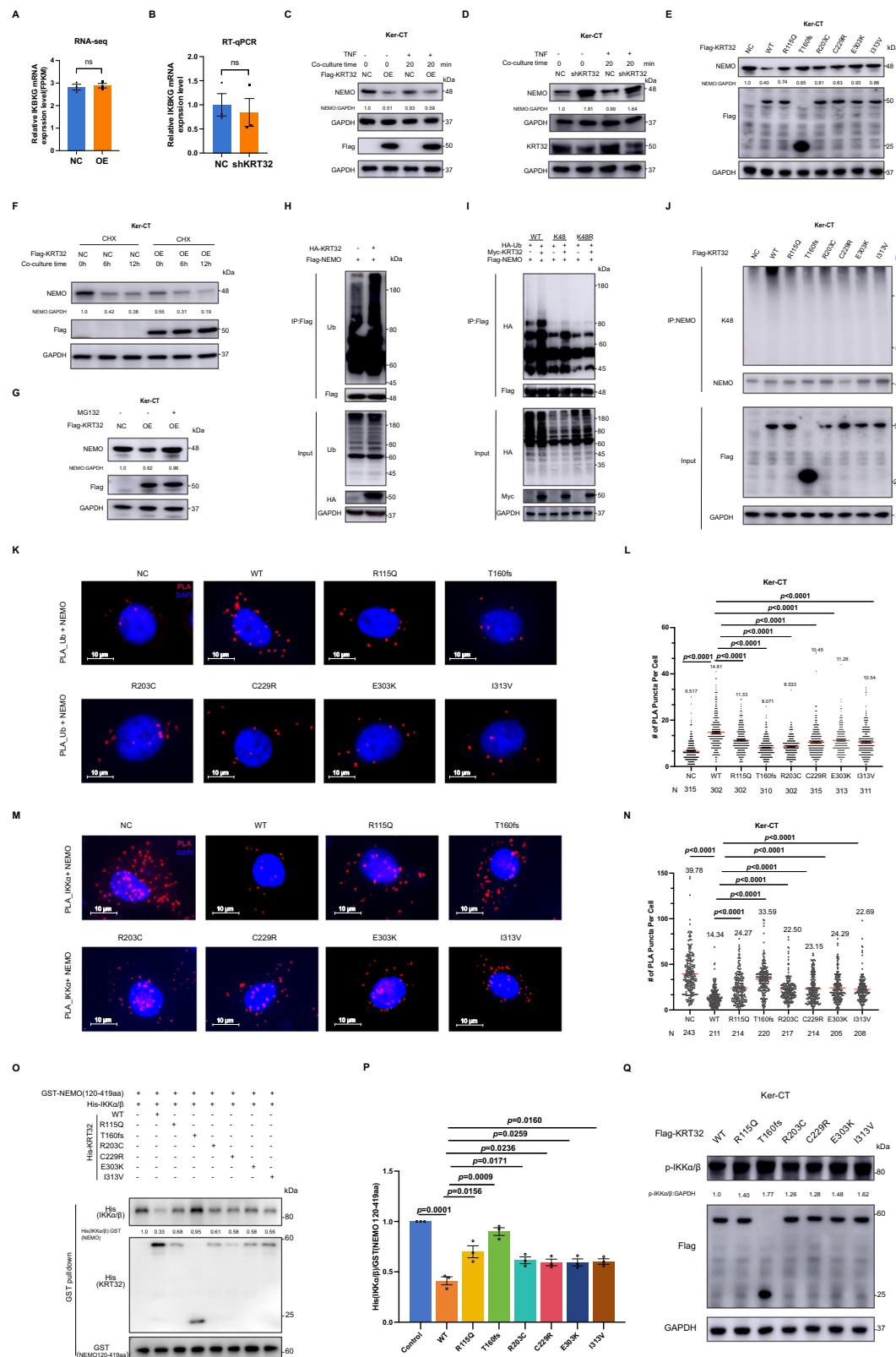

thicker skin and KRT32 mutation expression in the spinous layer. Our study indicates that KRT32 interacts with NEMO, inhibiting the activation of the NF-κB pathway to limit cellular proliferation. Therefore, KRT32 and/or the NF-κB pathway may serve as potential targets for PRP therapy.

KRT32 is comprised of three coil regions (Coil 1A, Coil 1B, and Coil 2) that constitute the IF rod domain, which is deemed as the functional domain of keratin. However, there exists limited information regarding the physiological function and protein structure of KRT32 as well as its involvement in pathological conditions. In the present study, we have identified 23 variants of the *KRT32* gene, out of which six were predicted to be pathogenic mutants. An in vitro analysis demonstrated that these six variants exhibited varying degrees of attenuation in inhibiting the NF-κB signaling pathway. Conversely, other common

**Fig. 7 | The interaction of KRT32 with NEMO promotes NEMO degradation via K48-linked polyubiquitination modification and also inhibits the formation of the IKK complex.** KRT32 expression level does not affect the transcription of IKBKG (the gene encoding NEMO) in Ker-CT cells. The mRNA level of NEMO in Ker-CT cells overexpressing KRT32 detected using RNA-seq (**A**) and in KRT32 knockdown cells detected by RT-qPCR analysis (**B**). Data are means ± SEM of three independent experiments in (**A**) and (**B**), and *P* value was calculated using a two-sided unpaired Student's *t* test. **C, D** Immunoblotting of NEMO in Ker-CT cells with overexpressing KRT32 and knockdown with and without TNF treatment. **E** The impact of KRT32 wildtype and mutations overexpression on NEMO protein level in Ker-CT cells analyzed by Western blot. **F** Cycloheximide (CHX) chase assay to analyze the protein stability of NEMO in Ker-CT cells overexpressing KRT32. Cells were treated with 50 μg/ml CHX for indicated time. **G** The levels of NEMO were measured by western blotting with 5 μM MG132 in Ker-CT cells overexpressing KRT32 wildtype or negative control for 12 h. **H** Co-immunoprecipitation of HA-KRT32 and Flag-NEMO in HEK 293T cells, followed by detection of ubiquitin levels of NEMO by Western blotting. **I** Co-transfection of Flag-NEMO with HA-Ub-WT, HA-Ub-K48 or K48R (containing lysine at residue K48 or lysine to arginine mutation at residue K48) along with Myc-KRT32 in HEK 293 T cells, followed by immunoprecipitation and Western blotting analysis. **J** Immunoprecipitation of lysates with NEMO antibody from Ker-CT cells overexpressing KRT32 wildtype or mutations, and then detect the Ub-K48 modification of NEMO. PLA analysis in Ker-CT cells overexpression KRT32 wildtype and mutations to assess the interaction of Ub with NEMO (**K**) and IKKα with NEMO (**M**). Scale bars represent 10 μm. **L, N** Quantification of the number of PLA foci per cell detected in (**K**) and (**M**), separately. Each dot on the graph corresponds to a specific analyzed cell. Red bars represent the mean ± SEM from the indicated number (*N*) of cells. The number of cells analyzed per group varies as follows: *N* = 315, 302, 302, 310, 302, 315, 313, 311 in (**L**) and *N* = 243, 211, 214, 220, 217, 214, 205, 208 in (**N**), with each group consisting of three biological replicates. *P* value was calculated using a two-sided unpaired Student's *t* test. **O** GST-NEMO (120-419aa) fusion protein was incubated with excess *E. coli* extracts containing His-KRT32 (wildtype or six mutants), His-IKKα, and His-IKKβ. GST complex was pulled down with glutathione-Sepharose beads, and the protein complexes were analyzed by western blotting. **P** Statistical analysis was performed on the binding ability of IKKα/β and NEMO with wildtype KRT32 and its mutations addition from tree-independent experiments in (**O**). Data shown as means ± SEM of three independent experiments in (**P**). *P* value was calculated using a two-sided unpaired Student's *t* test. **Q** Immunoblotting assay of phosphorylated IKKα/β in Ker-CT cells overexpressing KRT32 wildtype and mutations. One representative experiment from two independent experiments with similar results is shown in (**C**−**G**) and (**J**, **Q**). One representative experiment from three independent experiments with similar results is shown in (**O**). Source data are provided as Source Data file.

and benign variants did not significantly affect NF-κB pathway activation when compared to wild-type protein. Interestingly, all six mutations are localized within the IF rod domain of KRT32, resulting in a conformational change of said domain. Notably, the individual 2 frameshift mutation resulted in premature termination of protein translation. The truncated protein comprises a head region and partial IF rod domain with the tail region missing. Consequently, the functional defect of this mutation is the most apparent in the in vitro experiment. However, the truncated KRT32 protein still retains a partial inhibitory effect. To sum up, the anti-inflammatory effect of KRT32 may be closely related to its IF rod domain. The precise pathogenic mechanism requires further clarification.

Next, we validated the interaction between NEMO and KRT32. NEMO is the regulatory subunit of the inhibitor of kappaB kinase (IKK) complex that controls the activation of the NF-κB signaling pathway by phosphorylating the inhibitors of NF-κB[31], thus leading to the dissociation of the inhibitor-NF-κB complex and ultimately the degradation of the inhibitor[32,33]. The dysfunction of NEMO protein can lead to a variety of diseases, with incontinentia pigmenti (IP) being the prototypical syndrome associated with NEMO dysfunction that predominantly affects females[34]. Additionally, some patients with NEMO dysfunction exhibit hyperkeratosis, acanthotic epidermis, and follicular papules[31,34], which resemble the manifestations observed in patients with PRP. In the present study, we have identified six sites of *KRT32* mutants that directly or indirectly impair the interaction between KRT32 and NEMO, leading to abnormal activation of the NF-κB signaling pathway in skin keratinocytes. The interaction between KRT32 and NEMO promotes the K48 polyubiquitination modification of NEMO, resulting in its degradation. However, the specific E3 ubiquitination ligases recruited by KRT32 to facilitate the ubiquitination and degradation of NEMO require further exploration.

Moreover, the functional role of CARD14 in PRP pathogenesis has been revealed; however, in our present study, only one patient (1/58) was found to carry the *CARD14* mutation. Structurally, all CARD family proteins contain a CARD domain located at their amino-terminus[35]. The CARD domain is thought to function as an oligomerization domain that transduces the activation signal to the IKK complex through the C-terminal domain of BCL10, thus promoting binding and oligomerization of NEMO[36–38]. Although the precise mechanism by which CARD14 regulates NF-κB activation in PRP patients remains unclear, a separate study has demonstrated that CARD14 can modulate NEMO expression through RNF7-mediated regulation of NEMO ubiquitination, thereby affecting NF-κB signaling pathway activation[39]. These findings collectively suggest that pathogenic genes associated with PRP may directly or indirectly influence NEMO and contribute to disease onset.

It has been reported that microbial infection may act as a predisposing factor for PRP, leading to the upregulation of TNF expression and activation of skin inflammation[40–45]. In our study, we observed high levels of TNF in both serum and skin lesions of PRP patients. The healed PRP patient exhibits much lower serum TNF level compared to the serum TNF level in patients before treatment. And only following subcutaneous administration of TNF-induced PRP-like manifestation in the dorsal skin of *Krt32*[(−/−)] mice, suggesting a potential role for TNF in the pathogenesis of PRP. Meanwhile, the intrinsic *Tnf* transcription of keratinocyte cells in PRP mouse model increased in the process of TNF treatment compared to that before treatment and after restored (Supplementary Fig. 8F). Therefore, *Krt32* knockout may predispose individuals to PRP rather than spontaneous skin inflammation directly. The findings of vivo experiments indicate that the absence of KRT32 in mice could compromise the anti-inflammatory function of keratinocytes, rendering them less effective at defending against harmful external stimuli and leading to skin inflammation resembling PRP-like manifestations.

In summary, our study has identified six loss-of-function mutations in *KRT32* that predispose individuals to the development of PRP. These mutations hinder the ability of KRT32 to bind and inhibit NEMO expression, and then impair IKK complex formation. As a result, they trigger the upregulation of the NF-κB pathway in keratinocytes to cause immunoregulation imbalance. Our study has revealed the function of keratin family members in mediating inflammatory reactions and its role in the pathogenesis of PRP. The identification of NEMO as the common binding protein of KRT32 and CARD14 in the NF-κB signaling pathway provides a potential target for therapeutic intervention in PRP.

## Methods

### Ethics statement

The clinical investigations were conducted in accordance with the principles of the Declaration of Helsinki. Human ethics was approved by the ethical committee of the Shandong Provincial Institute of Dermatology and Venereology (2014KYKT23). All individuals who participated in the experiment were aware of the use of their samples and provided their written informed consent. Animal protocols used in the study were

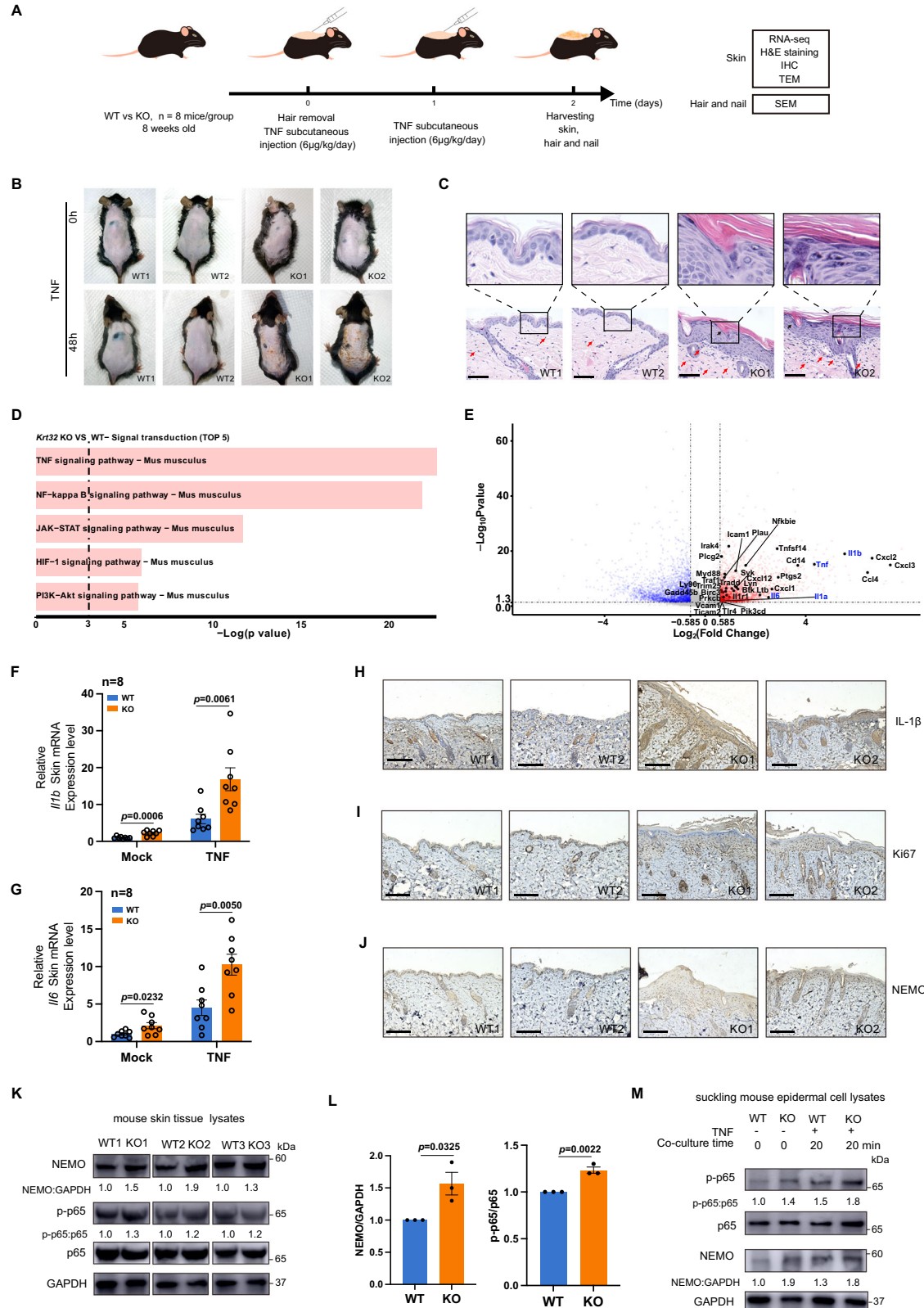

approved by the Animal Ethics Committee of Shandong Provincial Institute of Dermatology and Venereology (20210214DWLL001).

## Participants

From 2014 to 2021, all patients diagnosed with PRP at the Provincial Hospital for Skin Diseases, Shandong First Medical University, were recruited in the present study. The patients were collaboratively diagnosed by clinicians and pathologists, with clinical manifestations and histological examination results serving as the primary diagnostic criteria[6,46]. The detailed clinical information of all patients with PRP was presented in Supplementary Data 2, and summarized in Supplementary Table 2. Self-reported sex and ethnicity were recorded.

Healthy controls were recruited as volunteers from the regular physical examination center of the Shandong Provincial Hospital, with

**Fig. 8 | Skin phenotype in *Krt32* KO mice induced by TNF. A** Schematic representation of the experimental schedule. Eight-week-old *Krt32* wildtype (WT) and knockout (KO) C57BL/6J mice (*n* = 8 mice/group) were subcutaneously injected with TNF at a dose of 6 μg/kg/day body weight into their shaved dorsal skin for a duration of 48 h. The dorsal skin of *Krt32*(−/−) mice exhibited pronounced thickening and extensive yellow scaling, resembling the cutaneous manifestations observed in human patients with pityriasis rubra pilaris (PRP). Skin, hair, and nail samples were collected for RNA-seq, H&E staining, transmission electron microscopy (TEM), and scanning electron microscopy (SEM) analysis. **B**, **C** Two representative photos of the dorsal skin of WT and *Krt32* KO mice (*n* = 8 mice/group) treated by TNF, along with H&E staining showing lymphocytes and hair follicular plugging indicated by red and black arrows respectively. Results from another six mice are presented in Supplementary Fig. 8A, B. **D** Selected KEGG signal transduction pathways identified for significant DEGs *(P* < 0.05 and log₂FC > 0.585) through RNA sequencing of between *Krt32* WT and KO C57BL/6J mice (*n* = 8 mice/group) treated by TNF. The statistical test was hypergeometric test, and the level of significance was set at a two-sided *P* < 0.05 without multiple comparisons. **E** Selected NF-κB pathway-associated genes are highlighted. Colored points denote *P* < 0.05, which means −log

(*P*) > 1.30, dashed line), with red indicating upregulated genes (log₂FC > 0.585) and blue indicating downregulated genes (log₂FC < −0.585). The statistical test was Wald test, and the level of significance was set at a two-sided *P* < 0.05 without multiple comparisons. FC denotes fold change. **F**, **G** RT-qPCR analysis of *Il1b* and *Il6* expression in the epidermis of *Krt32*(−/−) mice (*n* = 8) and WT mice (*n* = 8) with pre- and post-TNF stimulation. Data are shown as means ± SEM in (**F**) and (**G**). *P* value was calculated using a two-sided unpaired Student's *t* test. Representative immunohistochemical staining of the dorsal skin of *Krt32* WT and KO mice treated with TNF for detecting IL-1β (**H**) Ki67 (**I**) and NEMO (**J**). Scale bar = 150 μm. Results from another six mice are presented in Supplementary Fig. 8C–E. **K** Western blot analysis of NEMO and p-p65 in the skin tissue lysates from *Krt32*(−/−) mice (*n* = 3) and WT mice (*n* = 3) with TNF treatment. **L** Statistical analysis the expression of NEMO and p-p65 in (**K**). Data are shown as means ± SEM in (**L**). *P* value was calculated using a two-sided unpaired Student's *t* test. **M** Western blot analysis of NEMO and p-p65 in epidermal cell lysates from a mixture of cells from four *Krt32* WT and KO suckling mice with and without TNF treatment. One representative experiment from two independent experiments with similar results is shown in (**M**). Source data are provided as Source Data file.

exclusion criteria applied to patients diagnosed with inflammatory and autoimmune diseases. Additionally, for the following functional experiments such as immunohistochemistry, immunofluorescence, RNA-seq, etc., we ensured matching of age, gender, and skin sampling site between healthy individuals and patients.

### Whole exome sequencing (WES) and Sanger sequencing
To perform WES in the discovery cohort, the TguideM16 Automatic Nucleic Acid Extractor (TIANGEN, Beijing, China) and the TGuide Large Volume Blood Genomic DNA Kit (OSR-M104, TIANGEN) were used to extract DNA from peripheral blood samples. The standardized genomic DNA was fragmented into random DNA fragments (150–200 bp) by using an ultrasonic crusher. Adenine molecules were added to the 3′-end of the DNA fragment, and Illumina connectors were added at both ends of the fragment for end repair to build a DNA library. The biotin-labeled RNA probe was fully mixed with the DNA library by using specific primers and hybridized in the liquid phase. Streptomycin-labeled magnetic beads were then captured and amplified by PCR to enrich the gene. FastQC v0.11.8 was used to assess the basic quality of the sequencing reads, which were then trimmed and filtered with Trimmomatic v0.38. Sequencing was performed on the Illumina NovaSeq 6000 platform. The resulting reads were mapped to the human genome reference hg19 by using the BWA v0.7.17 alignment tool, and the mapped reads were further filtered by GATK v4.0.12.0 software. All variants were annotated with ANNOVAR based on genome build hg19. The mutation frequency was estimated based on the dbSNP, HapMap, HGMD, and ExAC databases.

For the gene burden test, we included all rare damaging variants with a minor allele frequency (MAF) < 0.0005. Damaging variants were defined as those that cause nonsense, frameshift, or splice site variants or missense mutations predicted to be deleterious by four prediction algorithms including SIFT, PolyPhen-2 HumDiv, PolyPhen-2 HumVar, and Mutation Taster. After excluding the cases carrying the pathogenic *CARD14* variant, the remaining 16,026 damaging variants were subjected to the gene burden test. A gene-trait association was considered significant at $P < 2.50 \times 10^{-6}$ (0.05/20000, the gene number obtained after Quality Control).

For the validation cohort, Sanger sequencing was performed using the 3500xL Genetic Analyzer (Applied Biosystems, MA, USA).

### Cell culture and transfection
The immortalized keratinocyte cell line, Ker-CT, was obtained from the American Type Culture Collection (ATCC CRL-4048; Lot No. 70004879) and cultured in Complete medium KGM-Gold BulletKit (00192060, Lonza).

Primary human keratinocytes were isolated from children's foreskin surgical tissue obtained with written informed consent using protocols. Primary mouse keratinocytes were isolated from the neonatal mouse epidermis. Briefly, tissue samples were collected from surgical waste and sterilized with 75% ethanol for 30 min, followed by rinsing with sterilized PBS twice. The cleaned tissues were treated with 2.5 mg/ml dispase (17105041, Gibco) at 4 °C overnight. Subsequently, the epidermal layer was isolated from the tissue and digested by 0.05% trypsin for 10 min. Dulbecco's modified Eagle's medium (DMEM) with 10% fetal calf serum was used to stop the digestion and collect the cells and continue to culture these cells with DemaCult keratinocyte expansion Basal medium (100-0501, STEMCELL).

HEK293T cells were obtained from ATCC and cultured in DMEM medium supplemented with 10% (v/v) fetal bovine serum (N2813206P, Gibco). The cells were cultured in a humidified 5% CO₂ incubator at 37 °C. The KRT32 or other specific plasmids were transfected in HEK293T cells using Lipofectamine 2000 (11668019, Invitrogen) at a final concentration of 1 μg/mL following the manufacturer's protocol.

### Molecular structure simulations and docking
The AlphaFold program was utilized to obtain a rational three-dimensional structural model of KRT32 and NEMO proteins[47]. The stereochemical quality and accuracy of the protein model were evaluated and assessed through Ramachandran map calculations using the PROCHECK program after refinement.

To investigate the interaction patterns between KRT32 and NEMO, we performed molecular docking studies using the HDDock program. The structure with the highest docking score was selected as the standard result for subsequent interaction analysis. The docking score was computed/optimized using the iterative knowledge-based scoring function ITScorePP or ITScorePR. The confidence score was calculated as follows: Confidence_score = 1.0/[1.0 + e0.02*(Docking_Score+150)]. We defined confidence scores above 0.7 as a high probability of interaction between two molecules. All results were analyzed and visualized using PyMOL.

### Plasmid construction
*KRT32* (GenBank: NM_002278.3) cDNA was amplified from a human keratinocyte cDNA library. Specific primer pairs, Forward: 5′- ATGA-CATCCTCCTGCTGTGTCACC-3′ and Reverse: 5′- CAGTAGCGGCCCT GGGGGCAGGGTGAG-3′, were utilized for amplification of KRT32 mRNA and subsequent ligation to pFlag-CMV2, pMyc-CMV2, pHA-CMV2, pET-28a (+), and pGEX-6P-1 using a one-step cloning kit (Vazyme, Nanjing, China). *KRT32* point mutations primers (Supplementary Table 3) were designed using QuickChange Primer Design

Program, a web-based tool for primer design. *KRT32* point mutations were generated using standard molecular biology techniques.

## Dual-luciferase reporter assays

The assay was performed as previously reported[48]. Briefly, HEK-293T cells were seeded at a density of $1 \times 10^5$ cells per well in 24-well plates and cultured until they reached ~70–80% confluence. The cells were co-transfected with 500 ng of luciferase reporter (NF-κB-Luc) and 50 ng of pRL-TK renilla luciferase reporter, along with the appropriate control or expression plasmid (including KRT32 expression plasmid at a gradient of 100, 200, 400, and 800 ng or KRT32 WT and mutant expression plasmids predicted to be either damaging or benign variants based on their minor allele frequency (<0.0005 vs ≥0.0005)). At 24 h post-transfection, the cells were subsequently exposed to either 20 ng/mL TNF (P00029, Solarbio) or culture medium for a duration of 12 h. The dual-luciferase reporter assay was performed using the Dual-Luciferase Reporter Assay System (E1910, Promega) following the manufacturer's instructions. A fluorescence assay was conducted using the Cytation 5 imaging reader (Biotek, Vermont, USA). Firefly and Renilla luciferase activities were assessed and normalized with GraphPad Prism.

## Generation of lentivirus-mediated stable overexpression and knockdown cell lines

Stable overexpression of wild-type (WT) and mutations of KRT32 in the Ker-CT cell lines were constructed via lentivirus-mediated transfection, followed by selection with hygromycin (100 µg/ml, HY-K1051, MCE). And the corresponding empty vectors were used to generate control stable cell line. For stable knockdown cell line, pLKO.1-based shRNA against KRT32 were packaged via lentivirus, and then infect Ker-CT cells, followed by selection with hygromycin. Two shRNA sequences were used as follows, shRNA1: GTGTCACCAACAACTTGCA; shRNA2: GTGGAGGAATGGTTCAATA.

## Western blot assay

For the immunoblot assay, Ker-CT cells with either KRT32 over-expression or knockdown were treated with 20 ng/mL TNF for the indicated hours and subsequently lyzed in 100 µL RIPA buffer (R0010, Solarbio) supplemented with a protease and phosphatase inhibitor cocktail (P1049, Beyotime). The protein concentrations in the extracts were standardized and subsequently boiled with a protein loading buffer at 95 °C for 5 min. Equal amounts of protein from each group were separated by SDS-PAGE on a 10% gel, followed by transfer onto methanol-activated PVDF membranes (IPVH00010, Millipore). The membranes were blocked with TBST (TBS containing 0.1% Tween 20) containing 5% nonfat milk at room temperature for 1 h, and then incubated with the following antibodies for overnight at 4 °C: anti-Flag (1:1000; 14793; CST), anti-Flag (1:1000; 8146; CST), anti-KRT32 (1:1000; H00003882-B01P, Abnova), anti-KRT32 (1:1000; H00003882-D01P, Abnova), anti-HA (1:1000; 3724, CST) anti-p-IKKa/β (1:1000; 2697, CST), anti-p-p65 (1:1000; 3033, CST), anti-p65 (1:1000; 8242, CST), anti-NEMO (1:2000; 18474-1-AP, PROTEINTECH), anti-Ub (1:1000; 3936, CST), anti-K48 UB (1:1000; 4289, CST) and anti-GAPDH (1:20000; 60004-1-Ig, PROTEINTECH). Subsequently, the membranes were subjected to TBST washing and then incubated with HRP-conjugated secondary antibodies in a blocking buffer at room temperature for 1 h.

## Immunohistochemical analysis

Full-thickness skin tissues were collected and fixed in 10% buffered formalin, followed by embedding in paraffin wax, from PRP patients with *KRT32* mutations or PRP patients without *KRT32* mutation, normal controls, as well as *Krt32* knockout and wild-type mice. The paraffin-embedded tissues were sectioned at a thickness of 3 µm using a rotary microtome, and the sections were placed on a hot plate at 75 °C for

deparaffinization. Depending on the staining requirements, single-color staining was performed using the Two-Step Assay Kit (PV9000, ZSGB-BIO), or multiplex immunohistochemistry was conducted using a Multiplex immunohistochemistry kit (10001100100, panovue). The antibodies used in IHC analysis included: anti-TNF (1:100; BA0131, Boster), anti-IL-1β (1:100; sc-32294, Santa Cruz), anti-IL-6 (1:1000; ab9324, Abcam), anti-IL-8 (1:100; 27095-1-AP, PROTEINTECH), anti-Ki67 (1:100; A21861, abclonal), and anti-NEMO (1:100; 18474-1-AP, PROTEINTECH). The EVOS FL Auto 2 Imaging System (Thermo Fisher Scientific, MA, USA) and Vectra 3 Automated Quantitative Pathology Imaging System (Akoya Bio, MA, USA) were used to capture images of the sections.

## Cell proliferation assays

The stable KRT32 shRNA or KRT32 wildtype, mutants, and corresponding control overexpressing Ker-CT cells were seeded onto 96-well plates at the density of 2000 cells per well in a fresh medium for the CCK-8 assay. Cell viability was evaluated using the Cell Counting Kit-8 (CK04, Dojindo) at 24, 48, and 72 h. Following the addition of CCK-8 reagent, the microplates were incubated for an additional 2 h at 37 °C. The absorbance was measured at 450 nm using the Cytation 5 imaging reader (BioTek, CA, USA).

## 5-Ethynyl-2′-deoxyuridine staining

Cell proliferation was assessed using the 5-ethynyl-2′-deoxyuridine (EdU) assay with an EdU kit (C0075S, Beyotime). Ker-CT cells stably overexpressing KRT32 wildtype, mutants, and corresponding control were seeded onto a 48-well plate at 30% confluence during logarithmic growth and treated with the corresponding concentration of EdU reagent for 2 h. Subsequently, the cells were washed twice with PBS for 5 min and fixed in 4% paraformaldehyde at room temperature for 15 min, and then washing twice with 3% bovine serum albumin (BSA) in PBS. After permeabilization, the EdU click reaction was performed to fluorescently label the incorporated EdU within DNA, enabling visualization of proliferating cells. Then the cells were subsequently stained using Hoechst 33342 as a nuclear stain. The images were captured using the EVOS FL Auto2 Imaging System.

## RNA extraction, library preparation, and sequencing

For cell RNA-seq, cells were initially cultured for 24 h in a 12-well plate with a cell count of $10^6$. Total RNA was extracted using t the RNAiso Plus kit (TAKARA, Japan). For mice skin RNA-seq, the dorsal skin was separated using a scalpel (tissue size of 0.5 cm × 0.5 cm) and homogenized in RNA lysate reagent (LS1040, Promega) with the assistance of Gentle MACSTM Octo Dissociator (Miltenyi Biotec, Bergisch Gladbach, Germany). RNA quality was examined by gel electrophoresis and Qubit instrument (Thermo, Waltham, MA, USA). Sequencing was carried out using the Illumina Novaseq 6000 instrument by the commercial service of Genergy Biotechnology Co. Ltd. (Shanghai, China).

For FFPE RNA-seq, tissue samples from six patients with PRP carrying *KRT32* mutations and six age-, gender-, and site-matched healthy controls were obtained. FFPE tissue sections were processed for RNA isolation using Deparaffinization Reagent and CAT5 Reagent. The quality and purity of RNA were examined by a NanoDrop One/OneC spectrophotometer (Thermo Scientific, Waltham, MA, USA) and Life Invitrogen Qubit RNA BR (Broad-Range) Assay Kit. RNA integrity was analyzed using Agilent 4200 TapeStation system (Agilent, Santa Clara, CA, USA). The NEBNext Poly(A) mRNA Magnetic Isolation Module and NEBNext Ultra II mRNA Library Prep Kit for Illumina were used for mRNA isolation and library construction following the manufacturers' protocols. As for the quality control of library, Qubit dsDNA HS Assay Kit (Q32851, Thermo Scientific) was used to measure the concentration of library, then Agilent 4200 was used to examine the distribution of segments in library. Finally, library molar concentration was determined using the KAPA Library Quant kit (illumina) universal

qPCR Mix. And high-throughput transcriptome sequencing was performed on an Illumina NovaSeq6000 platform according to the manufacturer's instructions. The technical assistance is provided by Shanghai iProteome Biotechnology Co., Ltd (Shanghai, China).

The raw data was handled by Skewer and data quality was checked by FastQC v0.11.2. High-quality reads were aligned to the Human genome hg38 using STAR, StringTie. When the species is mice, clean reads were aligned to the Mouse genome mm10.

## Immunoprecipitation assay

For co-immunoprecipitation assays, HA-tagged KRT32 was co-transfected into HEK-293T cells with Flag-tagged IKKα, IKKβ, NEMO or IκBα plasmids for 24 h, respectively. Subsequently, the cells were lysed with Nonidet P-40 lysis buffer (AR0107, Boster) containing protease and phosphatase inhibitors. The lysates were subjected to co-IP using specific agarose-conjugated anti-DYKDDDDK-Tag antibodies (M20018, Abmart) for 2 h. After extensive washes, immunoprecipitated proteins were separated on SDS-PAGE, transferred to PVDF membranes and detected by Western blotting with appropriate antibodies.

For endogenous and semi-endogenous immunoprecipitation, cell lysates were lysed in Nonidet P-40 lysis buffer. Immunoprecipitation was carried out with Flag-conjugated agarose or anti-NEMO (1:50; 18474-1-AP, PROTEINTECH) antibody and protein Protein A/G-Agarose (A10001, Abmart), followed by overnight incubation at 4 °C. The immunocomplexes were extensively washed, separated by SDS–PAGE, and subjected to immunoblotting analysis using appropriate antibodies.

## Protein purification and pull-down assay

The plasmids encoding GST-tagged KRT32 (wild-type) and NEMO, as well as His-tagged KRT32 (wild-type and mutants), NEMO, IKKα, IKKβ, and IκBα were transformed into competent *Escherichia coli* BL21 (DE3) cells for 12 h, respectively. Subsequently, the cells were cultivated with 1 mM isopropyl beta-D-1-thiogalactopyranoside (I1020, Solarbio) at 16 °C for 12 h. The BL21 (DE3) material was then subjected to ultrasonic lysis and the fusion proteins were purified using the GST-tag Protein Purification Kit (P2262, Beyotime) and His-tag Protein Purification Kit (P2226, Beyotime), following the manufacturer's instructions.

For the pull-down assay, primary keratinocytes and Ker-CT cell lysate or excess His-tagged NEMO, KRT32 (wild type or 6 mutants), IKKα and IKKβ were incubated with GST-tagged KRT32 or NEMO fusion protein for 12 h. The GST complexes were subjected to the pull-down assay using glutathione-Sepharose beads, followed by western blotting analysis with anti-GST and anti-His antibodies to detect protein complexes.

## Identification of GST-KRT32 interacting proteins by mass spectrometry

To identify the KRT32-associated proteins, primary keratinocyte cell lysate was incubated with GST-tagged KRT32 and GST-tagged vector for 12 h ($n = 1$). The GST complexes were subjected to the pull-down assay using glutathione-Sepharose beads, enabling selective affinity purification. After this purification process, the enriched protein complexes were subjected to protein gel electrophoresis. Silver staining was performed to visualize the protein bands, which were then excised from the gel for further analysis. The excised gel bands were submitted to Jingjie PTM Biolab for mass spectrometry-based detection and analysis.

The tryptic peptides were dissolved in solvent A, directly loaded onto a homemade reversed-phase analytical column (25-cm length, 100 μm i.d.). The mobile phase consisted of solvent A (0.1% formic acid, 2% acetonitrile/in water) and solvent B (0.1% formic acid, 90% acetonitrile/in water). Peptides were separated with the following gradient: 0–22 min, 6–35% B; 22–26 min, 35–80% B; 26–30 min, 80% B,

and all at a constant flow rate of 550 nl/min on an EASY-nLC 1200 UPLC system (ThermoFisher Scientific). The separated peptides were analyzed in Orbitrap Exploris 480 (ThermoFisher Scientific) with a nano-electrospray ion source. The electrospray voltage applied was 2100 V. Precursors and fragments were analyzed at the Orbitrap detector. The full MS scan resolution was set to 60,000 for a scan range of 350–1800 m/z. Up to 20 most abundant precursors were then selected for further MS/MS analyses with 20 s dynamic exclusion. The HCD fragmentation was performed at a normalized collision energy (NCE) of 28%. Automatic gain control target was set at 50%, with an intensity threshold of 5000 ions/s and a maximum injection time of 200 ms.

The resulting MS/MS data were processed using Proteome Discoverer search engine (v2.4). Tandem mass spectra were searched against the human SwissProt database (20389 entries) concatenated with reverse decoy and contaminants database. Trypsin/P was specified as cleavage enzyme allowing up to two missing cleavages. Min. peptide length was set as 6. Mass error was set to 10 ppm for precursor ions and 0.02 Da for fragment ions. Carbamidomethyl on Cys was specified as fixed modification. Oxidation on Met and acetylation on protein N-terminal were specified as variable modification. False discovery rate was adjusted to <1%, minimum score for peptides was set >20 and peptide confidence was set high.

The mass spectrometry proteomics data have been deposited to the ProteomeXchange Consortium via the PRIDE partner repository with the dataset identifier PXD050141.

## Proximity ligation assay (PLA)

PLA assay was performed using Duolink PLA kit (DUO92008, Sigma-Aldrich) according to the manufacturer's instructions. For tissue paraffin sections, 3 μm sections were fixed in methanol, permeabilized with 0.15% Triton X-100, and blocked with the provided blocking buffer. Primary antibodies were applied and incubated overnight at 4 °C. For cell assays, cells were cultured on coverslips in 24-well plates overnight, fixed with 4% PFA, permeabilized with 0.15% Triton X-100, and blocked with 10% BSA. Primary antibodies were applied and incubated overnight at 4 °C. In both cases, after primary antibody incubation, coverslips were washed and then incubated with Duolink PLA probes (anti-mouse plus and anti-rabbit minus) for 1 h at 37 °C. Subsequent steps including ligation, amplification, and washing were performed according to the manufacturer's instructions. Finally, coverslips were counterstained with DAPI mounting medium, and images were acquired using a Vectra 3 Automated Quantitative Pathology Imaging System (Akoya Bio, MA, USA). The primary antibodies used for PLA analyses were: NEMO (1:200; 18474-1-AP, PROTEINTECH), KRT32 (1:100; H00003882-B01P, Abnova), Flag (1:100; 8146, CST), IKKα (1:1000; 11930, CST).

## Ubiquitination assays

For analysis of the ubiquitination of NEMO in HEK293T cells, cells were transfected with HA-tagged KRT32, Flag-tagged NEMO or HA-tagged Ub wild-type or Ub mutants. Whole-cell extracts were immunoprecipitated with agarose-conjugated anti-DYKDDDDK-Tag antibodies and analyzed by immunoblot with anti-Ub antibody (1:1000; SC-8017, Santa Cruz) or anti-HA antibody (1:1000; 3724, CST). For analysis of the ubiquitination of NEMO in Ker-CT cells overexpressing KRT32 wildtype or mutations, immunoprecipitation experiments were conducted using anti-NEMO antibodies in Ker-CT cells overexpressing KRT32 wild-type or mutations. Ubiquitination was detected using anti-K48 antibody (1:1000; 4289S, CST). Proximity ligation assay (PLA) experiments were performed to detect the binding of NEMO and Ub using anti-NEMO (1:100; SC-8032, Santa Cruz) and anti-Ub antibodies (1:100; ab7780, abcam).

## Cytokine detection through MSD assay

Cytokine levels in human serum and cell supernatant were quantified using V-PLEX Human Proinflammatory Panel II (4-Plex) (K15053D,

Article

MSD) on the MSD QuickPlex SQ120 platform according to the manufacturer's instructions.

## Mouse strains and generation of *KRT32* knockout mice

C57BL6/J background mice were used in our study. The standard laboratory chow diet for mice was purchased from Beijing Keao Xieli Feed Co., Ltd. (Cat# SPF-CXSL). The SPF-grade animal room was maintained with a humidity level of 45–60% and a 12-h light/dark cycle (8:00 a.m.–8:00 p.m.). The female and male mice of 8 weeks mice were euthanized by cervical dislocation at the end of the experiment, after which dorsal skin was separated for further analysis. All animal experiments were approved by the Animal Ethics Committee of Shandong Provincial Institute of Dermatology and Venereology.

*Krt32* knockout C57BL/6J mice were generated by Cyagen Biosciences. Briefly, CRISPR/Cas9-mediated genome engineering through electroporation was used to knock out the *Krt32* gene of C57BL6/J background mice. gRNA1 and gRNA2, which respectively match the forward strand of the gene with sequences GGCAAGGTCCGTGGCAACACAGG and GGCTGGCCAAACCAACTATCAGG, were introduced into fertilized eggs via high-throughput electroporation. Homozygous knockout mice *Krt32*[(−/−)] and their corresponding control littermates *Krt32*[(+/+)] were generated by breeding of heterozygous *Krt32*[(+/−)] mice.

## Real-time quantitative reverse transcription-polymerase chain reaction (RT-qPCR)

Eight-week-old *Krt32* wildtype (WT) and knockout (KO) C57BL/6J mice ($n = 8$ mice/group) were subcutaneously injected with and without TNF (6 μg/kg/d) on their shaved dorsal skin for 48 h. Subsequently, the dorsal skin was separated using a scalpel and homogenized in RNA lysate reagent (LS1040, Promega) with the assistance of Gentle MACS™ Octo Dissociator (Miltenyi Biotec, Bergisch Gladbach, Germany). Total RNA was extracted using the Eastep Super Total RNA Extraction Kit (LS1040, Promega). The reverse transcriptase reaction was performed using 1 μg of total RNA in a 20 μl reaction volume using FastKing gDNA Dispelling RT SuperMix (KR118, TIANGEN). Subsequently, amplification was carried out using 20 ng of cDNA and forward and reverse primers at a concentration of 400 nM each. The specific primer sequences used are listed in Supplementary Table 4. The geometric mean values of Actb were utilized for normalization, while the relative transcription level was assessed using the ΔCt method. Real-time polymerase chain reaction reactions were performed in triplicate with iTaq Universal SYBR Green SuperMix (Bio-Rad, California, USA) on Step One Plus thermocycler (Applied Biosystem, MA, USA).

## Scanning electron microscopy (SEM)

Hair and nail samples were collected from both *Krt32*[(−/−)] and wild-type mice. Following treatment with TNF, which induced skin phenotypes similar to PRP in the *Krt32*[(−/−)] mice, samples were prepared for SEM analysis. Prior to imaging, all samples were coated with a thin layer of gold using a sputter coater (ETD-2000, Beijing Elaborate Technology Development Ltd.). Imaging was performed using a COXEM EM-30 Plus SEM microscope (Coxem, Daejeon, Korea).

## Transmission electron microscopy (TEM)

TEM was used to confirm and analyze the quantification of desmosomes and hemidesmosomes. After *Krt32*[(−/−)] mice treated with TNF exhibited skin phenotypes similar to PRP, skin tissue samples (0.5 mm × 0.5 mm × 2 mm) from both *Krt32*[(−/−)] and wild-type control mice were immediately fixed in 2.5% glutaraldehyde overnight at 4 °C. The skin tissues were washed with PBS three times and fixed in 1% osmium tetroxide for 2 h at room temperature. Following dehydration through a graded series of ethanol (50%, 70%, 90%, and 100%), the samples were embedded in Epon resin. Sections (70 nm) were prepared and stained with uranyl acetate and lead citrate after the blocks

were cured at 60 °C for 48 h. Finally, the sections were analyzed and imaged under a TEM (JEOL-1200, Weiya Bio Co., Ltd.). Imaging focused on capturing the basal lamina hemidesmosomes and the intercellular desmosomes of keratinocytes within the same layer. For each skin sample from both *Krt32*[(−/−)] and wild-type mice, at least 20 images were systematically captured to ensure a robust statistical analysis. The collected data were then subjected to quantitative analysis using Student's *T* test to assess the significance of observed differences.

## Statistical analysis

A two-sided Fisher's exact test was performed using SPSS software v26. A gene-trait association was considered significant at $P < 2.50 \times 10^{-6}$ (0.05/20000, the gene number obtained after quality control) for all damaging variants.

All other data are presented as mean ± standard error of mean (SEM), and were analyzed by two-tailed paired/unpaired Student's *t* test, one way ANOVA, two-ANOVA with Dunnett's multiple comparisons test or Pearson correlation coefficient, using Prism v8.3 (GraphPad). Statistical significance was considered at $P < 0.05$. No statistical method was used to predetermine the sample size. No data were excluded from the analyses.

## Reporting summary

Further information on research design is available in the Nature Portfolio Reporting Summary linked to this article.

## Data availability

The RNA-seq data from *Krt32* knockout and wildtype mice dermal response to TNF stimulation in this study have been deposited in the Genome Sequence Archive (GSA) database under accession code CRA015401. The RNA-seq data of KRT32 overexpressing Ker-CT cell lines, and FFPE samples from 6 PRP with *KRT32* mutations in this study have been deposited in the Genome Sequence Archive (GSA) for Human database under accession code HRA006924 and HRA006901 respectively. The WES data of PRP identifies rare mutations in KRT32 in this study and have been deposited in the Genome Sequence Archive (GSA) for the Human database under accession code HRA005276. The protein data regarding the interaction of GST-KRT32 identified through Pulldown-MS in this study has been deposited in the ProteomeXchange Consortium under accession code PXD050141. The genetic data obtained from participants in this project have been approved by the China Human Genetic Resources Management Office. All data generated or used during the study appear in the submitted article and its Supplementary Information or are available from the corresponding author upon request. Uncropped and unprocessed scans of blots have been provided as in the Source data file. Source data are provided in this paper.

## Code availability

Data analysis was performed with publicly available packages, as described in the Methods. No custom code was generated in this study.

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

## Acknowledgements

This study was funded by grants from the Central guidance for local scientific and technological development projects of Shandong

Province (YDZX2023058, H.L.), the Key research and development program of Shandong Province (2021LCZX07, F.R.Z.), Shandong Provincial Natural Science Foundation (ZR2021QC047, P.D.S.) and National Natural Science Foundation of China (82201956, P.D.S.). We are grateful to the patients and healthy controls for donating skin tissue or blood.

## Author contributions

H.L. and F.R.Z. designed this study. Data curation and formal Analysis: Z.Z.W., X.X.L., P.D.S., W.C.L., and W.J.C.; Investigation and performed experiments: P.D.S., W.J.C., F.M.J., C.Z.Z., T.T.L., C.W., Y.Z., Z.H.M., Y.H.S., X.C.C., S.L.C., G.Z.Z., Y.X.L., Y.J.L., F.X.B., and Q.S.; Visualization: P.D.S., W.J.C., and Z.Z.W.; Writing-original draft: P.D.S., W.J.C. and H.L.; Writing-review & editing: P.D.S., X.X.L., H.L., J.J.L., Q.Y., M.O.O., and F.R.Z.

## Competing interests

The authors declare no competing interests.

## Additional information

[1]Hospital for Skin Diseases, Shandong First Medical University, Jinan, Shandong, China. [2]Shandong Provincial Institute of Dermatology and Venereology, Shandong Academy of Medical Sciences, Jinan, Shandong, China. [3]Department of Dermatology, Qilu Hospital, Shandong University, Jinan, Shandong, China. [4]Department of Pharmacology and Therapeutics, University of Liverpool, Liverpool, UK. [5]Genome Institute of Singapore, Singapore, Singapore. [6]School of Public Health, Shandong First Medical University and Shandong Academy of Medical Sciences, Jinan, Shandong, China. [7]Shandong University of Traditional Chinese Medicine, Jinan, Shandong, China. [8]These authors contributed equally: Peidian Shi, Wenjie Chen. ✉e-mail: hongyue2519@hotmail.com; zhangfuren@hotmail.com

