## [Peer Review File · Nature Communications]

Loss-of-function mutations in Keratin 32 gene disrupt skin immune homeostasis in pityriasis rubra pilarisEditorial note: Parts of this Peer Review File have been redacted as indicated to maintain patient confidentiality.

REVIEWER COMMENTS

Reviewer #1 GWAS/WES, gene function in skin (Remarks to the Author):

In this manuscript, the authors revealed that loss-of-function mutations in KRT32 gene could cause PRP. Mechanistically, they demonstrated using KRT32 disrupted keratinocytes that mutated KRT32 could not bind to NEMO, leaving its insufficient degradation, that led to excessive activation of NF- κ B signaling pathway. Finally, the skin lesions in KRT32 KO mice recapitulated the pathologic features of PRP. They provide significant findings of the role for mutated KRT32 in the development of PRP by a sophisticated proof through clinical to experimental procedures including mouse model. There are some points to be addressed.

1. Supplementary Fig.2: While the serum TNF levels in PRP patients are elevated, it would be helpful to show them in patients after treatment, or them with spontaneously healed skin disease. In other words, how about any correlation of the serum TNF levels with disease severity?
2. Supplementary Fig.3: Immunohistochemical staining for inflammatory cytokines are needed in the uninvolved skins from patients with KRT32 mutation.
3. Supplementary Fig.4D: Immunostainings for KRT32 in PRP patients without KRT32 mutations are missing.
4. KRT32 in the skin lesions of RPR patients with mutations is expressed in the supra-basal. They should describe this.
5. Figure 1G, H: Cytokine levels of WT and NC are so different. They should explain this.
6. Data of TNF-untreated group should be shown in Fig.G, H.
7. Fig. 6D, E, F: Western blot bands of interest should be quantified with a densitometric analysis.
8. Fig. 6J: They mentioned in the text that “six damaging variants of KRT32 overexpressed in HEK293T cells impair the expression of KRT32 itself, leading to upregulation of NEMO protein expression, which correspondingly led to higher level of phosphorylation of

IKKa/IKKb compared to the KRT32 wild-type protein” could be read that alteration of NEMO was simply due to low expression of KRT32. This notion is inconsistent with the finding of Supplementary Fig.4D, in which KRT32 is highly expressed in the epidermis of PRP patients with the same mutations, although its distribution is altered. The expression of NEMO in p.Cys229Arg is increased in the basal layer, however, where KRT32 expression in the same patients is almost absent. Therefore, please show figures of double staining of NEMO and KRT32 in patients with/without KRT32 mutations compared to healthy controls.

9. Supplementary Fig.7: Electron micrograph images of KO mice. Were they treated with TNFa? They should compare them between in TNF-treated and untreated mice.

10. KRT32, a type I acidic keratin, heterodimerizes with basic type II keratins (KRT82?), and is involved in the structural integrity of keratinocyte cytoskeleton, therefore, they should describe if any abnormality in the hair, nails, in clinical or histopathological manifestations specific for PRP patients with KRT32 mutations compared with those without mutations. Also, they should examine in detail if any phenotype of nails and hair of KO mice following treatment with TNFa.

11. Figure 8B: This reviewer has an interest in the hair follicle plugging-like change in KO mice, which resembles the typical PRP histological feature.

Reviewer #2 gene polymorphisms, skin inflammation (Remarks to the Author):

This manuscript by Shi and colleagues examines the association of mutations in keratin 32 and PRP. The authors perform whole exome sequencing and a range of biochemical and murine tests to determine if K32 plays a role in PRP. Though the data are somewhat interesting the manuscript contains multiple significant issues that need to be resolved.

The experiments are not rigorous and do not show clearly the impact of heterozygous K32 mutations on promoting PRP in families is not convincing.

1. I find it striking that none of the PRP cases you identified in your Han Chinese patient population between 2014-21 had a family history of PFP; how thoroughly did you examine their familial medical histories? This was not stated. How could these patients have a germline mutation and be the only person in their families with PRP. This indicates that the

mutation is not a strong driver of PRP.

2. What clinical lesions did the patients have, you need a table showing what clinical features the PRP patients demonstrated. Was nail pitting examined?

3. Did any of your patients have psoriasis-some of the histologic sections look like psoriasis.

4. Fig 2. You need to provide H+E stained images of these samples , some of them look like psoriasis not PRP. NC and NC2 do not look like adequate control epidermis, are they matched for site, sex and age with the PRP samples. Again, the presentation of the data is suboptimal and difficult to interpret, high magnification and corresponding H+E images would help.

5. Fig 2 and 3. Were the K32 overexpression experiments done with wildtype K32 or a mutant form, it is not clear. If wildtype K32 was used in the experiment and this influences HaCaT cell proliferation than the mutations are of unclear biological significance. Were any mutant form of K32 studied in this way? Was there a difference between WT and mutant K32 in these assays?

6. Fig 3, HEK293 cells are inappropriate for characterizing the biological effects of K32 in keratinocytes. Use primary human keratinocytes or NHEK cells. HaCaT cells have p53 mutations not appropriate for the studies described. Again, primary human keratinocytes or NHEK cells.

7. Fig 4. Many proteins bind to keratins, did you show differential binding to WT K32 versus mutant forms. I did not find this data. The data in 6K are not clear.

8 Fig 6K. Data shown are not clear if the difference between the WT and mutants is significant than why not plot the data and show it?

Reviewer #3 pityriasis rubra pilaris, skin inflammation (Remarks to the Author):

Peidian Shi and colleagues identify novel mutations in Keratin 32 (KRT32) gene in patients

with pityriasis rubra pilaris (PRP), and provide evidence that loss of function mutations in this gene impair the ability of KRT32 protein to regulate NF- κ B and IL-1 signalling which leads to the hyper-activation of the NF κ B-IL-1-pathway, leading to skin inflammation. The authors provide data suggesting that KRT32 can interact with NEMO and that this binding promotes NEMO-ubiquitination and degradation, which in turn inhibits NF κ B-activation. In addition, KRT32 is shown to regulate the expression of an IL-1R antagonist and an IL-1 receptor decoy (IL-1R2). Of note, the authors present data showing that the knockout of Krt32 in mice leads to an inflammatory skin phenotype resembling PRP.

The key findings of this study are the identification of novel loss-of-function mutations in the KRT32 gene in patients with pityriasis rubra pilaris (PRP), and evidence that these mutations lead to dysregulation of NF- κ B and IL-1 inflammatory signalling in keratinocytes.

Overall the data are novel and interesting. However, some of the conclusions were performed using cell lines that are not relevant to the studied disease the statistical power of the mouse experiments appears to be limited. Additionally, it would be important to verify that the mouse model adequately recapitulates human PRP.

In summary, the study proposes KRT32 mutations as an important genetic factor in PRP pathogenesis and identifies a novel role for KRT32 in regulating skin inflammation. The results provide novel insights into the pathogenesis of the disease that are likely to be of interest to the dermatology, immunology and keratin biology fields.

Questions and suggestions

1. to verify that Krt32 KO mice adequately recapitulate human PRP, transcriptomic alterations in KRT32 KO mice and in PRP patient skin should be compared, not only a few selected genes.

2. Supplementary Figure 4D: Immunohistochemistry of KRT32 expression in skin tissues obtained from PRP patients with KRT32 mutations, PRP patients without KRT32 mutations shows altered localization of the protein within the epidermis. While KRT32 appears to localize to basal cell layer in normal skin, its localization, but not the expression level, seems to be altered in PRP patients where it appears to be expressed in the spinous cell layers.

Please explain and discuss. This important difference and its potential implications need to be described and discussed in the manuscript. What controls were used to ensure the specificity of the antibodies used?

3. to assess whether KRT32 overexpression affects the studied pathways through NEMO or through alternative mechanisms rescue experiments are needed. Does knockdown/knockout of NEMO restore normal phenotype in KRT32-mutant cells? Does NEMO-knockdown/knockout results in observations that are comparable to those obtained with KRT32 -overexpression?

4. the reliance on HaCaT cells in all in vitro experiments is one of the major limitations of this work. HaCaT is a genetically abnormal, hypotetraploid cell lines with p53 and many other mutations. While findings from HaCaT cells can provide preliminary insights, making definitive conclusions about normal keratinocyte biology and disease mechanisms requires more physiologically relevant cell models. I recommend the authors address this limitation by performing key experiments such as: a) Assessing impact of KRT32 knockout/overexpression on NF- κ B activity and cytokine production in primary keratinocytes or immortalized keratinocytes (such as TERT-immortalized keratinocytes), b) Evaluating binding of KRT32 mutants to NEMO in primary or immortalized keratinocytes, c) Testing effects of KRT32 modulation on proliferation and inflammation in patient-derived PRP keratinocytes if available. In any case the authors need to discuss the limitations of the HaCaT model and interpreting results from these cells cautiously.

5. The interaction between the NEMO and KRT32 was confirmed in HEK293T cells (in which KRT32 is not expressed under physiological conditions) and in cell-free systems. Considering the central place of this interaction in the model proposed by the authors, the interaction needs to be investigated using primary keratinocytes/epidermal lysates by co-IP. Can KRT32 pulled down by anti-NEMO Abs in keratinocytes?

6. The authors claim that KRT32 promotes the K48-linked polyubiquitination-mediated degradation of NEMO protein and impedes the assembly of IKK complex (Figure 6). However, as far as this reviewer can tell, all experiments were performed in HEK293T cells. To validate the physiological relevance of KRT32-NEMO-interaction these results need to be confirmed in a keratinocyte-based system, especially considering that KRT32 does not seem to be expressed physiologically in HEK293T cells and its overexpression may result in non-physiological interactions.

7. Description of the methods related to the treatment of KRT32 KO mice with TNF- α is missing. Information about the route, total number and frequency of administration, should be clearly described in the methods section, ideally with a graphical summary in Figure 8.

Controls for the efficient treatment in WT and KO mice with TNF alpha are needed (i.e. the measurement of TNF-response genes before and after treatment).

8. qPCR analysis was performed on three mice only, this is way too low to draw statistically valid conclusions considering the variability among mice. This number needs to be increased. On how many mice were the experiments presented on Figure 8A-D performed? Considering the variability among individual mice here, too the number should significantly more than two or three.

9. Additional experiemnts are needed are needed to evaluate the effect of Krt32 mutation on inflammation and NEMO-expression in the mouse epidermis. WB from epidermal cell lysates could be used to compare changes in NEMO between WT and KO mice. Which cells in the mouse epidermis express Krt32 protein? Is NEMO-expression altered in the corresponding cell layer in KO mice? Is the localization of p65 altered in the mutant mice?

10. Transcriptomic analysis of TNF- α -stimulated HaCat cells with overexpression of KRT32 was performed, however results seem to be shown only for four transcripts. The results from these experiments should be provided in the form of supplementary tables along with false-discovery-rate adjusted p-values and fold changes. Was the IL-1 pathway significantly enriched among the differentially expressed genes? What was the results of unbiased analysis, such as GO-enrichment analysis on this data set? Was NFKB-pathway affected? Were there any other processes altered, too? Results from a more detailed analysis need to be presented and discussed. Why did the authors focus on IL1A/B, and IL1Rs in this analysis?

11. The analysis of inflammatory cytokine levels in PRP skin was performed by IHC, which is a semi-quantitative method. How were the results quantified? Data obtained with quantitative method needs to provided (e.g. RNAseq, qRT-PCR).

12. The figure legend for Supplementary Figure 4D seems to be incorrect.

“Immunohistochemistry of KRT32 expression in skin tissues obtained from PRP patients with KRT32 mutations, PRP patients without KRT32 mutations, and healthy individuals” -however IF images are shown only for heathy individuals and patients with KRT32 mutations.

13. The authors claim that direct immunofluorescence assays demonstrated colocalization of KRT32 and NEMO in HaCaT cells (Figure 4). However, these are low-magnification IF-images which are not conclusive in terms of co-localization. Proximity ligation assay, or super-resolution microscopy would be need with quantitative image analysis to support this claim. Again, this result was obtained from HaCaT cells. Are KRT32 and NEMO colocalized in

primary keratinocytes?

14. Figure 3C: it is surprising that treatment of keratinocytes with TNF does not result in increased p65-phosphorylation – Is this a technicality due to the use of HaCaT cells or is this an experimental variation? It seems to contradict to panel E, where nuclear p65 is increased at the same time point.

15. Figure 7F: western blot data should be shown about the expression of KRT32 in these conditions. I find it very surprising that overexpression of K32 could completely block IL-1b-induced p65-phosphorylation. What is the authors explanation for this? What is the baseline Krt32-expression in these cells? IL1R signals through several pathways, can KRT32 inhibit all of them?

16. I wonder whether alterations in cell-to-cell or cell-to-ECM junctions are affected in KRT32-mutant patients and whether signalling from the junctional complexes (e.g. via catenins) could mediate the observed effects. Does loss-of-function of KRT32, similar to that of other intermediate filaments, impact desmosomal, hemi-desmosomal interactions? Did the authors see any signs of this in their histological and EM- analyses? Please discuss this and if appropriate provide additional images covering areas of the basal lamina and cell junctions in Supplementary Figure 7.

17. How was the specificity of the antibodies used for the immunohistochemistry analysis of IL-1R2 and IL-1Ra confirmed? The images provided show mainly cytoplasmic and, in some cases, even nuclear expression for them. Is this the expected staining pattern for these proteins in normal epidermis?

18. The heritability of PRP is non-mendelian, thus heterozygous KRT32 mutations are unlikely to be the sole reason for the disease. The authors should discuss other factors that would be needed to trigger disease onset in combination with KRT32 mutations.

REVIEWER COMMENTS

Reviewer #1 GWAS/WES, gene function in skin (Remarks to the Author):

In this manuscript, the authors revealed that loss-of-function mutations in KRT32 gene could cause PRP. Mechanistically, they demonstrated using KRT32 disrupted keratinocytes that mutated KRT32 could not bind to NEMO, leaving its insufficient degradation, that led to excessive activation of NF- κ B signaling pathway. Finally, the skin lesions in KRT32 KO mice recapitulated the pathologic features of PRP. They provide significant findings of the role for mutated KRT32 in the development of PRP by a sophisticated proof through clinical to experimental procedures including mouse model. There are some points to be addressed.

1. Supplementary Fig.2: While the serum TNF levels in PRP patients are elevated, it would be helpful to show them in patients after treatment, or them with spontaneously healed skin disease. In other words, how about any correlation of the serum TNF levels with disease severity?

Reply: Thank you for your comment. We detected the TNF level in serum from the healed patients after treatment, and found that the healed PRP patients exhibited lower serum TNF level compared with those before treatment (**Supplementary Figure 3D**). Referring to an established individual Pityriasis Rubra Pilaris area and severity index (iPRPASI) (Shao et al, ADR, 2023, PMID: 37532946), the serum TNF level indicates a positively correlation with disease severity of PRP patients (Pearson correlation 0.7273, p-value <0.0001) (**Supplementary Figure 3E**). Furthermore, in PRP mouse model, the expression of TNF in the skin also shows a significant increase during the disease progression compared to that before treatment and after restored (**Supplementary Figure 8F**). Thus, the serum TNF level might be considered as a potential biomarker for the extent of diseases severity and the recovery of PRP patients. We have now included these points in the revised manuscript.

Supplementary Figure 3D and 3E

(D) The secretion of TNF- α in the serum of PRP patients with and without *KRT32* mutations was

measured before treatment and healed (n = 9). P value was calculated using a two-sided unpaired Student's t test. (E) Correlation between serum TNF- α levels with PRP area and severity index (iPRPASI) scores in the study group (n = 47) were analyzed by Pearson correlation analysis (Pearson's $r = 0.7273$, $p < 0.0001$).

F

Supplementary Figure 8F

(F) The *Tnf α* transcription in the skin of *Krt32* KO mice with non-treatment, treatment and restored from TNF- α treatment (n = 8).

2. Supplementary Fig.3: Immunohistochemical staining for inflammatory cytokines are needed in the uninvolved skins from patients with KRT32 mutation.

Reply: Thank you for your suggestion. We regret the inability to collect samples from the uninvolved skins of patients. To address this limitation, we conducted the following two experiments:

a) We meticulously compared the expression levels of inflammatory cytokines in 6 PRP patients with *KRT32* mutations and 6 healthy controls, matched for age, gender, and site with PRP patients, using transcriptomic analysis and immunohistochemical staining. This comparison revealed elevated expression of inflammatory cytokines in the skin lesions of patients compared to those of healthy individuals, as illustrated in the revised **Supplementary Figure 3A and 3B**, and **Supplementary Figure 4**.

b) To comprehensively assess the data across the body, we measured *Tnf* levels in both damaged and uninvolved skins using RT-qPCR in the PRP mouse model. The results indicated a significant decrease in the transcriptional level of *Tnf* in uninvolved skins (see **figure below**). Therefore, we concluded that inflammatory cytokines are predominantly expressed in the damaged skin region of PRP patients and contribute to the progression of PRP disease.

The mRNA levels of *Tnf* in the lesional and uninvolved skins were quantified by RT-qPCR in PRP mouse model (n=8).

3. Supplementary Fig.4D: Immunostainings for KRT32 in PRP patients without KRT32 mutations are missing.

Reply: We sincerely apologize for our oversight. Immunostainings for KRT32 in PRP patients without *KRT32* mutations have now been included in **Supplementary Figure 1D**. We re-performed the immunostainings with the matched healthy control, PRP patients with and without *KRT32* mutations.

Supplementary Figure 1D

(D) Immunohistochemical analysis of KRT32 expression in skin tissues obtained from PRP patients with *KRT32* mutations, PRP patients without *KRT32* mutations (n = 6), and healthy controls (n = 6). Age, gender, and skin sampling site matching were ensured between healthy individuals and patients. Scale bar = 150 μ m. The red dotted line denotes the epidermal-dermal boundary.

4. KRT32 in the skin lesions of RPR patients with mutations is expressed in the supra-

basal. They should describe this.

Reply: Thank you for your suggestion. IHC assay indicates the mutant *KRT32* of PRP patients mainly expressed in both basal and supra-basal layer, different from these in healthy controls and in PRP patients without *KRT32* mutations (**Supplementary Figure 1D**). We detailed describe it in the manuscript as followings:

“Additionally, our immunohistochemical (IHC) analysis revealed distinct patterns of mutant KRT32 of PRP patients compared to that in healthy controls and PRP patients without KRT32 mutations (Supplementary Figure 1D). In health controls, KRT32 was predominantly expressed in the basal keratinocytes. However, in PRP patients with KRT32 mutations, the epidermal layer was thickened, and KRT32 showed uniform expression in both basal and supra-basal keratinocytes. Interestingly, in PRP patients without KRT32 mutations, although the epidermal layer also became thickened, KRT32 was prominently expressed in basal keratinocytes rather than supra-basal keratinocytes.” (Line 136-145).

5. Figure 1G, H: Cytokine levels of WT and NC are so different. They should explain this.

Reply: Thank you for pointing this out. The difference of cytokines levels of WT and NC might be caused by several reasons: 1) Different methods to construct cell lines. For Figure 2G, we transiently transfected CRISPR-Cas9 mediated *KRT32* knockout plasmid (KO) and its corresponding control plasmid (WT) into HaCat cells by electroporation; As for Figure 2H, we used lentivirus-mediated stably overexpression of *KRT32* (OE) and its control vector (NC). Thus, the different background of engineered cell lines might lead to different cytokines expression in response to TNF- α stimulation. 2) The approach of direct quantitative method. We directly measured the concentration of cytokines. Thus, the cytokines expression can be affected by the cell confluency and/or cell status in the different batches of experiments.

To address this concern, we further re-performed these experiments using Ker-CT cells (as reviewer 2 comments #6 requested) with lentivirus-mediated stably overexpression and knockdown of *KRT32*. To make sure the consistency of data, we elaborately unified the same cell number when seeding cells and the accurate TNF- α simulating time in different batches of experiments. Three biological repeats were conducted to make sure the accuracy of results in these experiments. Now the cytokines levels of the two control samples become similar and new data are presented in revised **Figure 4E-4G** and **Figure 4J-4L**.

Figure 4E-4G and Figure 4J-4L

(E-G) Detection of IL-1 β , IL-6, IL-8 secretion in supernatant of KRT32 knockdown Ker-CT cells with or without TNF- α treatment by MSD assay. (J-L) MSD assay detection of IL-1 β , IL-6, IL-8 secretion of Ker-CT cells overexpressing KRT32.

6. Data of TNF-untreated group should be shown in Fig G, H.

Reply: Thank you for this suggestion, we have included the TNF-untreated group in the revised **Figure 4E-4G and Figure 4J-4L** and showed that TNF- α significantly induced the expression of inflammatory cytokines.

7. Fig. 6D, E, F: Western blot bands of interest should be quantified with a densitometric analysis.

Reply: We re-performed the experiments in Fig. 6D-F with Ker-CT cells and quantified the relevant western blot bands with densitometric analysis using ImageJ software. The signal intensity of NEMO was normalized to GAPDH, and relative values were labelled as shown in **Figure 7C, 7D, 7G**.

Figure 7C, 7D, 7G

(C, D) Immunoblotting of NEMO in Ker-CT cells with overexpressing KRT32 and knockdown with and without TNF- α treatment. (G) The levels of NEMO were measured by western blotting with 5 μ M MG132 in Ker-CT cells overexpressing KRT32 wildtype or negative control for 12 hours.

8. Fig. 6J: They mentioned in the text that “six damaging variants of KRT32 overexpressed in HEK293T cells impair the expression of KRT32 itself, leading to upregulation of NEMO protein expression, which correspondingly led to higher level of phosphorylation of IKK α /IKK β compared to the KRT32 wild-type protein” could be read that alteration of NEMO was simply due to low expression of KRT32. This notion is inconsistent with the finding of Supplementary Fig.4D, in which KRT32 is highly expressed in the epidermis of PRP patients with the same mutations, although its distribution is altered. The expression of NEMO in p.Cys229Arg is increased in the basal layer, however, where KRT32 expression in the same patients is almost absent. Therefore, please show figures of double staining of NEMO and KRT32 in patients with/without KRT32 mutations compared to healthy controls.

Reply: We apologize for the confusion. We have conducted double immunofluorescence staining of NEMO and KRT32 in PRP patients with *KRT32* mutations, comparing them to healthy controls and PRP patients without *KRT32* mutations. The *KRT32* damaging mutations were found to localize in both the basal and supra-basal layers, unlike *KRT32* WT, which mainly localizes in the basal layer, as shown in **Supplementary Figure 6**. Additionally, the expression of NEMO was significantly higher in PRP patients with *KRT32* mutations compared to healthy controls and PRP patients without mutations. This finding is consistent with our previous results indicating that *KRT32* damaging mutations reduce their interaction with NEMO and limit ubiquitin-mediated proteasomal degradation of NEMO in cells, as depicted in **Figure 7**.

Supplementary Figure 6. Double immunofluorescent staining of NEMO and KRT32 of the lesioned skins from PRP patients with/without *KRT32* mutations

Skin tissue sections obtained from healthy controls and PRP patients with and without *KRT32* mutations were immunostained to detect NEMO (red) and KRT32 (green). Scale bars represent 150 μm . Representative immunohistochemical images of NEMO and KRT32 staining from two healthy individuals and six cases of PRP with and without *KRT32* mutations are present.

Given that HEK293T does not express KRT32 itself, HEK293T is considered as an inappropriate cell line to characterize the biological effects of KRT32 as mentioned by reviewer 2 comment#6. Now we use Ker-CT cells to address the functions of KRT32 in the revised manuscript, and found that the damaging mutations in the keratinocyte cells would not impair the expression of KRT32 itself, which is more consistent with KRT32 expression in these patients (**Figure 7E**).

Furthermore, cellular experiments shed light on the mechanisms underlying how these

damaging mutations lead to attenuated affinity between KRT32 and NEMO, resulting in reduced NEMO ubiquitination and stabilization of NEMO protein (see **Figure 7E-7L**).

9. Supplementary Fig.7: Electron micrograph images of KO mice. Were they treated with TNF α ? They should compare them between in TNF-treated and untreated mice.

Reply: We are sorry for the vague description here. These samples were indeed from the skin samples treated with TNF- α . Following your suggestion, we compared the overall keratinocyte cell phenotypes between in TNF-treated and untreated mice by electron microscopy, and did not find significant alterations in morphological structures of keratinocytes in the skin of PRP mouse model. We have incorporated the additional TNF untreated results in **Supplementary Figure 10A** and updated the corresponding sections of the manuscript (**lines 337-341**) to reflect these findings accurately.

Supplementary Figure 10A

The overall keratinocyte cell phenotypes in *Krt32*^(-/-) and wildtype mice observed with and without TNF treatment by electron microscopy. Scale bar = 5 μ m.

10. KRT32, a type I acidic keratin, heterodimerizes with basic type II keratins (KRT82?), and is involved in the structural integrity of keratinocyte cytoskeleton, therefore, they should describe if any abnormality in the hair, nails, in clinical or histopathological manifestations specific for PRP patients with KRT32 mutations

compared with those without mutations. Also, they should examine in detail if any phenotype of nails and hair of KO mice following treatment with TNFa.

Reply: We carefully examined all the clinical manifestations of the 102 PRP patients, including the 6 patients with damaging *KRT32* mutations (**Supplementary Table 3**). The results revealed that 24 out of 102 patients exhibited nail involvement. Among the 6 patients with *KRT32* mutations, only one individual (Patient 5 with *KRT32* c.907G>A mutation) exhibited nail involvement. Additionally, only 3.9% of 102 PRP patients exhibited hair loss, and none of them belonged to the 6 patients with damaging *KRT32* mutations (**Supplementary Table 4**). And we did not find any other specific clinical manifestation for PRP patients with *KRT32* mutations compared with those without mutations.

Additionally, we did not observe any significant differences in the appearance of the nails and hair of *Krt32* knockout mice compared to WT mice following TNF- α treatment with the naked eye. Subsequently, we conducted microscopic observations of the skin, hair, and nails. We first analyzed the structural integrity of keratinocyte cytoskeleton by electron microscopy. Neither the desmosomes nor the hemidesmosomes revealed significant in mock and PRP mouse model (**Supplementary Figure 10B-E**). And also, analysis by scanning electron microscope indicated normal surface morphology of the hair from *Krt32* knockout and WT mice (see **Supplementary Figure 9A**). However, we observed a notably rough surface of the nails in *Krt32* knockout mice, as depicted in **Supplementary Figure 9B**.

Following your suggestions, we conducted Co-IP experiments to test the interaction of KRT32 and KRT82, indicating that KRT32 does not interact with KRT82 (**refer to the figure below**). Besides, in our pulldown assay experiments, KRT82 protein was not found in our list of KRT32-interacting proteins (see **Supplementary Table 9**). Thus, the current evidence demonstrated that KRT32 does not directly affect the structural integrity of keratinocyte cytoskeleton.

CoIP analysis of the interaction of KRT32 and KRT82. The lysates of Ker-CT cells overexpressing Flag tagged KRT32 were prepared, immunoprecipitated using anti-Flag antibodies, and subjected to an immunoblotting assay to detect the co-immunoprecipitated KRT32 proteins.

Supplementary Figure 10B-E

(B, C) The representative images of hemidesmosomes and desmosomes of keratinocytes from PRP mouse model. Scale bar = 1 μ m. Hemidesmosomes (arrows in yellow); Desmosomes (arrows in red)

(D) The number of hemidesmosomes per μ m. At least 13 fields for each sample from different mice were quantified. (E) The length of individual desmosome, >100 desmosomes were quantified for each sample from different mice. Data are means \pm SEM, and P value was calculated using a two-sided unpaired Student's t test (ns: no significant difference).

Supplementary Figure 9. SEM observation of hair and nail surfaces in *Krt32* wild-type and KO mice.

Hair samples from the backs (A) and nail samples from the hind limbs (B) of *Krt32* wild-type and knockout (KO) mice treated with TNF α (n = 4 mice/group) were observed using scanning electron

microscopy (SEM). Scale bar = 10 μ m. Boxed areas are amplified in inserts to indicate roughness on the surface of nails.

11. Figure 8B: This reviewer has an interest in the hair follicle plugging-like change in KO mice, which resembles the typical PRP histological feature.

Reply: Thanks. We indeed noted this feature in both the HE-stained samples of patients and in the *Krt32* KO mouse model. We have included detailed descriptions and discussions of these observations in the revised manuscript (**Lines 305-307**). Specifically, **Figures 1** and **8C** now highlight the hair follicular plugging, as indicated by black arrows.

Reviewer #2 gene polymorphisms, skin inflammation (Remarks to the Author):

This manuscript by Shi and colleagues examines the association of mutations in keratin 32 and PRP. The authors perform whole exome sequencing and a range of biochemical and murine tests to determine if K32 plays a role in PRP. Though the data are somewhat interesting the manuscript contains multiple significant issues that need to be resolved.

The experiments are not rigorous and do not show clearly the impact of heterozygous K32 mutations on promoting PRP in families is not convincing.

1. I find it striking that none of the PRP cases you identified in your Han Chinese patient population between 2014-21 had a family history of PRP; how thoroughly did you examine their familial medical histories? This was not stated. How could these patients have a germline mutation and be the only person in their families with PRP. This indicates that the mutation is not a strong driver of PRP.

Reply: Thank you for your feedback. To validate the family histories of PRP patients, we cross-referenced their information with medical records. No family history was noted. Additionally, we conducted follow-up via telephone calls and on-site visits the past three months. However, neither the patients nor their parents confirmed a family history of PRP. Only two patients reported a family history of psoriasis, which couldn't be confirmed as the family members currently have no lesions. We double-checked the clinical manifestations and histopathological features of the two patients, indicating PRP (**see the figures below**).

[figure redacted]

Clinical and histopathological features of two family history of psoriasis patients

Histological analysis of skin biopsy samples revealed characteristic PRP pathology, including layered parakeratosis, alternating hyperkeratosis and parakeratosis, hair follicular plugging.

Among the five subtypes of PRP, only Type V (atypical juvenile type) has been reported to have familial clusters, indicating a genetic predisposition (Telem et al, Am J Hum Genet, 2012, PMID: 22703878; Takeichi et al, JAMA Dermatol 2017, PMID: 27760266). Type V patients typically manifest symptoms in childhood and constitute only 5% of all PRP cases. In our cohort, nine patients developed PRP in childhood. However, four should be classified as Type III PRP, and the remaining five as Type IV PRP based on clinical characteristics. And the two PRP patients aforementioned who reported family history of psoriasis developed PRP in 62 and 48 years old, respectively.

Our findings and other studies suggest that PRP may result from a complex interplay between genetic and environmental factors. We propose that carrying the damaging mutations of *KRT32* significantly increases the risk of PRP, but its occurrence also

requires the confluence of other environmental factors, such as bacterial and viral infections, drugs, among others (Joshi et al, Am J Clin Dermatol, 2024, PMID: 38159213). Our animal model studies show PRP-like manifestations only in the dorsal skin of *Krt32*^(-/-) mice after subcutaneous TNF- α injection. This suggests that KRT32 knockout may predispose individuals to PRP rather than inducing spontaneous skin inflammation directly. Therefore, we agree with the reviewer that the mutations in KRT32 will increase the risk for developing PRP, instead of being a strong driver of PRP. Further discussion on this topic is available in our manuscript (see **lines 406-420**).

2. What clinical lesions did the patients have, you need a table showing what clinical features the PRP patients demonstrated. Was nail pitting examined?

Reply: Thank you for your feedback. We have compiled the clinical manifestations of 102 PRP patients, including detailed characteristics of the 6 patients with deleterious mutations, and all this information has been incorporated into **Supplementary Table 3** and **Supplementary Table 4**.

Indeed, we examined the patients' nails. Among them, 24 showed nail involvement, presenting with distal yellow-brown discoloration, subungual hyperkeratosis, nail thickening, and splinter hemorrhages. Among the 6 patients with *KRT32* mutations, only one exhibited nail involvement, as indicated in **Supplementary Table 3**. However, we did not observe typical nail pitting in these patients.

3. Did any of your patients have psoriasis-some of the histologic sections look like psoriasis.

Reply: While it's true that psoriasis and PRP share certain pathological similarities, such as epidermal thickening, hyperkeratosis, parakeratosis, and a perivascular mononuclear infiltrate in the dermis, there are distinct features of PRP evident in the high-resolution H&E staining images of these patients:

- ① Focal incomplete keratinization, alternating hyperkeratosis, and parakeratosis, without Munro microabscesses.
- ② Presence of the granular layer.
- ③ Irregular thickening of the spinous layer, with no epidermal thinning above the dermal papillae.
- ④ Sparse lymphoid cell infiltration around superficial dermal blood vessels, without neutrophilic infiltration.
- ⑤ Hair follicle plugging-like change.

All these histopathological features of PRP in our study were reconfirmed by two pathologists. We have provided clear clinical and histopathological images of six patients carrying deleterious mutations in **Figure 1** for reference.

4. Fig 2. You need to provide H+E stained images of these samples, some of them look like psoriasis not PRP. NC and NC2 do not look like adequate control epidermis, are they matched for site, sex and age with the PRP samples. Again, the presentation of the data is suboptimal and difficult to interpret, high magnification and corresponding H+E images would help.

Reply: Thank you for your suggestions. The high magnification and corresponding H+E images are now included in the revised **Figure 1**. We have carefully reviewed the clinical and pathological characteristics of patients in this study, and they meet the diagnosis criteria of PRP. We re-collected the skin samples from the healthy controls to make sure the site, sex, and age matching with those from respective PRP patients with *KRT32* mutations (**refer to the table below**). We re-matched NC1 and NC2, and added four matched healthy individuals to more clearly show the influence of *KRT32* damaging mutations on cell proliferation using the immunofluorescence staining of Ki67 (in revised **Figure 3A**).

[figure redacted]

Figure 1. Clinical and histopathological features of six PRP patients with *KRT32* mutations

Photographs of cutaneous lesions and histological analysis of skin biopsy specimens from PRP patients with *KRT32* mutations are presented. These patients manifested typical pathological characteristics of PRP, including widespread erythematous plaque coalescence, layered parakeratosis, absence of Munro microabscesses, alternating hyperkeratosis and parakeratosis, irregular thickening of the stratum spinosum, sparse lymphoid cell infiltration around superficial dermal blood vessels, and hair follicular plugging (labelled by black arrows). Each group of H&E-stained histological images are captured at two magnification levels – 100× (left) and 200× (right), and the mutation site is presented below the respective images.

Skin Sample Matching: PRP patients vs. Controls

With KRT32 mutation					Without KRT32 mutation				Healthy controls			
individual					PRP				HC			
No.	sex	age (y)	site	mutations	No.	sex	age (y)	site	No.	sex	age (y)	site
1	Male	53	Upper Limb	c.344G>A(p.Arg115Gln)	PRP1	Male	50	Upper Limb	HC1	Male	55	Upper Limb
2	Male	76	Chest	c.477_478del(p.Thr160fs)	PRP2	Male	73	Chest	HC2	Male	70	Chest
3	Female	41	Waist	c.607C>T(p.Arg203Cys)	PRP3	Female	43	Waist	HC3	Female	38	Waist
4	Female	36	Chest	c.685T>C(p.Cys229Arg)	PRP4	Female	40	Chest	HC4	Female	33	Chest
5	Male	67	Abdomen	c.907G>A(p.Glu303Lys)	PRP5	Male	62	Abdomen	HC5	Male	60	Abdomen
6	Female	54	Abdomen	c.937A>G(p.Ile313Val)	PRP6	Female	56	Abdomen	HC6	Female	53	Abdomen

Figure 3A

Representative immunofluorescence staining for Ki-67 of skin biopsy sections obtained from patients with *KRT32* mutation and matched healthy controls (HC; n = 6). Scale bar represent 150 μ m. Age, gender, and skin sampling site matching were ensured between healthy individuals and patients.

5. Fig 2 and 3. Were the K32 overexpression experiments done with wildtype K32 or a mutant form, it is not clear. If wildtype K32 was used in the experiment and this influences HaCaT cell proliferation than the mutations are of unclear biological significance. Were any mutant form of K32 studied in this way? Was there a difference between WT and mutant K32 in these assays?

Reply: Thank you for your comments. In Fig 2 and Fig 3, we indeed overexpressed wildtype *KRT32*. In the manuscript, we aimed to determine the gain- or loss-of -functions of the damaging mutations of *KRT32*. Following your suggestion, we overexpressed *KRT32* WT and all the six damaging mutations in Ker-CT cell, indicating that *KRT32* WT suppresses the cell proliferation but the damaging *KRT32* mutations lose such suppressing ability as measured by both of the CCK8 and EdU fluorescence labelling assays (in revised **Figure 3B-3E**).

As for the functions of *KRT32* on activation of NF- κ B, expression of *KRT32* mutations show an increase in the phosphorylated p65 and inflammatory cytokine compared to WT (**Figure 4M- 4O**). Meanwhile, we also detected multiple other features of *KRT32* mutants, including the distinct binding to NEMO (**Figure 6D** and **6E**), ubiquitin modification (**Figure 7J-7L**), regulation of NEMO protein homeostasis (**Figure 7E**) and also the IKK complex formation (**Figure 7M-7P**).

Therefore, all the results indicate that the PRP with damaging *KRT32* mutations result in a loss-of-function in promoting in cellular proliferation and NF- κ B activation.

Figure 3B-3E

(B, C) CCK-8 assays to assess the function of KRT32 on cell proliferation through KRT32 knockdown (B) or wildtype (WT) and six KRT32 mutations overexpression (C) in Ker-CT cells. shKRT32a and shKRT32 represent two different shRNA sequences against KRT32. Data shown as means \pm SEM of four repeats in B and C. P value was calculated using a one-way ANOVA in B and C. (D, E) EdU assay to evaluate the influence of KRT32 mutations on Ker-CT cell proliferation. One representative experiment shown in D. Scale bars represent 75 μ m. Statistical analysis of EdU-positive cells in three independent biological repeats (E).

6. Fig 3, HEK293 cells are inappropriate for characterizing the biological effects of K32 in keratinocytes. Use primary human keratinocytes of NHEK cells. HaCaT cells have p53 mutations not appropriate for the studies described. Again, primary human keratinocytes or NHEK cells.

Reply: As suggested by you and reviewer#3 comments #4, we used primary human keratinocytes cells and Ker-CT cell line to re-perform these and other experiments. Similar to the result from HaCat cell line, KRT32 knockdown increases the p65 phosphorylation and cytokine production in Ker-CT cells and overexpression decreases the NF- κ B activation in both primary human keratinocytes and Ker-CT cell line (in revised **Figure 4D-4L**).

We still left the only experiment NF- κ B-Luc assay using 293T cell line in the revised manuscript. Even if we have tried our best to optimize the experimental conditions to transfect our luciferase reporter (NF- κ B-Luc) and pRL-TK renilla luciferase reporter plasmids into primary human keratinocytes or Ker-CT cell, the transfection efficiency is still too low to meet the requirements. Since 293T owns a complete NF- κ B signaling pathway, it is a well-recognized tool to build a NF- κ B dependent luciferase reporter

system in many other cells (Voyer et al, Nature, 2023, PMID: 37938781; DengNC, et al, Nat Commun, 2018, PMID: 30385786). And The 293T-NF-kappaB dependent luciferase reporter system has been also used in the study of keratinocytes (Zhang et al, J Invest Dermatol, 2015, PMID: 25501661; Nickoloff et al, Cell Death Differ, 2002, PMID: 12107827). Thus, we used the 293T cells just as a tool to evaluate the capacity of KRT32 mutations in the activation of NF-κB pathway. Furthermore, we also used other methods to validate the roles of KRT32 in NF-κB activation, including western blot and MSD in primary human keratinocytes or Ker-CT cell. The revised images are now included in the revised **Figure 4**.

Figure 4D-4L

(D) The phosphorylated p65 expression in KRT32 knockdown Ker-CT cells with and without TNF-α (20 ng/mL) treatment for the indicated time. (E-G) Detection of IL-1β, IL-6, IL-8 secretion in supernatant of KRT32 knockdown Ker-CT cells with or without TNF-α treatment by MSD assay. (H-I) The phosphorylated p65 expression in primary keratinocytes and Ker-CT cells with Flag-tagged KRT32 overexpression. (J-L) MSD assay detection of IL-1β, IL-6, IL-8 secretion of Ker-CT cells overexpressing KRT32.

7. Fig 4. Many proteins bind to keratins, did you show differential binding to WT K32 versus mutant forms. I did not find this data. The data in 6K are not clear.

Reply: We did not map the differential interacting proteomic profiles of KRT32 WT/Mutants proteins in cells. Since in the assay of GST-tagged wildtype KRT32 pulldown from the cell lysis, NEMO ranks the first score in the list of proteins interacting with KRT32 (included in **Supplementary Table 9**). The abnormal NF-κB signaling pathway has been documented as a pivotal role in PRP disease. In our study, we identified the closely relationship between damaging KRT32 mutations and NF-κB signaling pathway by RNA-seq from FFPE tissues and NF-κB luciferase reporter assay. And no other proteins in NF-κB signaling pathway exist in this list. Thus, NEMO is the main target of KRT32 to regulate the activation of NF-κB.

To clearly elucidate the interaction between KRT32 and NEMO in Fig 6K, we further investigate the interaction of KRT32 and its mutant forms with NEMO and evaluate the activation of NF- κ B using different methods. Proximity ligation assay (PLA) experiments in Ker-CT cells are illustrated in **Figures 6D** and **6E**. We conducted PLA experiments to quantify effects of KRT32 compete with IKK α to reduce the complex formation of IKK-NEMO in **Figures 7M** and **7N**. Additionally, we have revised and clarified the presentation of data in the previous Figure 6K, which is now represented as **Figure 7O** in the updated version of our manuscript.

Figure 6D and 6E

(D) PLA analysis of the interaction of Flag-tagged KRT32 wildtype and mutations with NEMO in Ker-CT cells. Scale bars represent 10 μ m. (E) Quantification of the number of PLA foci per cell in D. Each dot on the graph corresponds to a specific analyzed cell. At least 100 cells were analyzed per experiment. Red bars represent the mean \pm SEM from the indicated N number of cells analyzed from two independent experiments. P value was calculated using a two-sided unpaired Student's t test (****p < 0.0001).

Figure 7M and 7N

(M) PLA analysis in Ker-CT cells overexpression KRT32 wildtype and mutations to assess the interaction of IKK α with NEMO. Scale bars represent 10 μ m. (N) Quantification of the number of PLA foci (red) per cell detected in M. Each dot on the graph corresponds to a specific analyzed cell. At least 100 cells were analyzed per experiment. Red bars represent the mean \pm SEM from the indicated N number of cells analyzed from at least two independent experiments.

8. Fig 6K. Data shown are not clear if the difference between the WT and mutants is significant than why not plot the data and show it?

Reply: We have quantified the protein bands previously presented in Figure 6K using ImageJ software and plot them in a bar graph. The updated data are now represented as **Figure 7O and 7P** in our manuscript, with numerical values provided under the bands to clearly depict protein levels.

Additionally, we have conducted proximity ligation assay (PLA) experiments to further evaluate the impact of wildtype and mutant KRT32 on IKK complex formation in cells, indicating that KRT32 variants lose the ability to compete with IKK α to bind NEMO, as demonstrated in **Figures 7M and 7N**. These complementary experiments enhanced the robustness of our study's findings and provide additional supporting of the differential effects of wildtype and mutant KRT32 on IKK complex assembly.

Figure 7O and 7P

(O) GST-NEMO (120-419aa) fusion protein was incubated with excess *E. coli* extracts containing His-KRT32 (wildtype or six mutants), His-IKK α , and His-IKK β . GST complex was pulled down with glutathione-Sepharose beads, and the protein complexes were analyzed by western blotting. (P) Statistical analysis was performed on the binding ability of IKK α / β and NEMO with wildtype KRT32 and its mutations addition from three independent experiments of O.

Reviewer #3 pityriasis rubra pilaris, skin inflammation (Remarks to the Author):

Peidian Shi and colleagues identify novel mutations in Keratin 32 (KRT32) gene in patients with pityriasis rubra pilaris (PRP), and provide evidence that loss of function mutations in this gene impair the ability of KRT32 protein to regulate NF- κ B and IL-1 signalling which leads to the hyper-activation of the NF κ B-IL-1-pathway, leading to skin inflammation. The authors provide data suggesting that KRT32 can interact with NEMO and that this binding promotes NEMO-ubiquitination and degradation, which in turn inhibits NF κ B-activation. In addition, KRT32 is shown to regulate the expression of an IL-1R antagonist and an IL-1 receptor decoy (IL-1R2). Of note, the authors present data showing that the knockout of *Krt32* in mice leads to an inflammatory skin phenotype resembling PRP.

The key findings of this study are the identification of novel loss-of-function mutations in the KRT32 gene in patients with pityriasis rubra pilaris (PRP), and evidence that these mutations lead to dysregulation of NF- κ B and IL-1 inflammatory signalling in keratinocytes.

Overall the data are novel and interesting. However, some of the conclusions were performed using cell lines that are not relevant to the studied disease the statistical power of the mouse experiments appears to be limited. Additionally, it would be important to verify that the mouse model adequately recapitulates human PRP.

In summary, the study proposes KRT32 mutations as an important genetic factor in PRP pathogenesis and identifies a novel role for KRT32 in regulating skin inflammation. The results provide novel insights into the pathogenesis of the disease that are likely to be of interest to the dermatology, immunology and keratin biology fields.

Questions and suggestions

1. to verify that *Krt32* KO mice adequately recapitulate human PRP, transcriptomic alterations in KRT32 KO mice and in PRP patient skin should be compared, not only a few selected genes.

Reply: Thank you for the suggestion. Since there is no fresh tissue for RNA-seq, we performed RNA-seq on paraffin-embedded tissue sections from PRP patients with *KRT32* mutations and matched healthy controls (**Supplementary Figure 3A and 3B**). Additionally, we conducted RNA-seq on the skin samples from *Krt32* KO mice after TNF- α treatment as for PRP mouse model and Ker-CT cells with KRT32 overexpression (**Figure 8D and 8E; Figure 4A and 4B**). The enriched pathways by KEGG analysis in *Krt32* KO mice could adequately recapitulate human PRP patients (**refer to figure below**), especially the gene expression changes in NF- κ B and TNF pathways as we described in this manuscript. Transcriptome of Ker-CT with KRT32 overexpression also indicates the suppressing of gene expression in NF- κ B pathway. Thus, all the transcriptomes show the consistency of PRP patients, mouse and cell model.

KEGG pathway analysis of RNA-Seq data from PFFE of PRP patients with *KRT32* mutations, , *KRT32*-overexpressing Ker-CT cells and TNF- α -treated *Krt32* KO mouse skin (PRP mouse model).

2. Supplementary Figure 4D: Immunohistochemistry of KRT32 expression in skin tissues obtained from PRP patients with KRT32 mutations, PRP patients without KRT32 mutations shows altered localization of the protein within the epidermis. While KRT32 appears to localize to basal cell layer in normal skin, its localization, but not the expression level, seems to be altered in PRP patients where it appears to be expressed in the spinous cell layers. Please explain and discuss. This important difference and its potential implications need to be described and discussed in the manuscript. What controls were used to ensure the specificity of the antibodies used?

Reply: Thank you for comments. We re-performed the immunostainings with the matched healthy control, PRP patients with and without *KRT32* mutations (as requested by reviewer 2 comment #4) (in the revised **Supplementary Figure 1D**). *KRT32* is predominantly expressed in the basal layer, and weakly expressed in spinous cell layer in healthy control, which is consistent with the single cell type expression data of *KRT32* from the HPA and GTEx database (**Supplementary Figure 1A-1C**). While in PRP patients with *KRT32* mutations, the epidermal layer was thickened and the *KRT32* shows a similar expression in both basal and spinous keratinocytes. However, the skin of PRP patients without *KRT32* mutation also show a thick epidermal layer, *KRT32* still remains high expression in basal keratinocytes, rather than spinous keratinocytes. Therefore, we believed that the damaging *KRT32* mutations change their localization in skin tissues. We described this point in the Result part and included the detailed discussion of *KRT32* location in Discussion part (**Lines 351 – 360**).

“Basal keratinocytes serve as mitotically active progenitor cells, gradually differentiating into cells of the upper layers. We hypothesize that KRT32 in the basal layer regulates skin cell proliferation and differentiation, with loss-of-function mutations of KRT32 in PRP patients inhibiting basal cell differentiation and maintaining strong proliferation in the spinous layer. Consequently, PRP patients present with thicker skin and KRT32 mutation expression in the spinous layer. Our study indicates that KRT32 interacts with NEMO, inhibiting the activation of the NF-κB pathway to limit cellular proliferation. Therefore, KRT32 and/or the NF-κB pathway may serve as potential targets for PRP therapy.”

KRT32 antibody we used here is from Abnova (H00003882-B01P). To validated the specificity of *KRT32* antibody, we used this antibody to probe *KRT32* protein in the *KRT32* knockdown Ker-CT cell and results show that the specific band decreases in both the two cell lines (**Supplementary Figures 2C**). And then we probed the Flag-*KRT32* in stably expressed Flag-*KRT32* of Ker-CT cells, which showing that the signal bands of Flag antibody and *KRT32* antibody are merged at the exactly same position (**Supplementary Figures 2B**). All the results confirmed the specificity of this antibody.

Supplementary Figures 2B and C

The efficacy of *KRT32* wildtype and mutant overexpression in Ker-CT cells was validated through Western blot analysis using anti-Flag monoclonal antibody.(B)The efficiency of *KRT32* knockdown

in Ker-CT cells was verified by Western blot analysis. Two different shRNAs (shKRT32a and shKRT32) were used here.

Supplementary Figure 1D

(D) Immunohistochemical analysis of KRT32 expression in skin tissues obtained from PRP patients with *KRT32* mutations, PRP patients without *KRT32* mutations ($n = 6$), and healthy controls ($n = 6$). Age, gender, and skin sampling site matching were ensured between healthy individuals and patients. Scale bar = 150 μm . The red dotted line denotes the epidermal-dermal boundary.

3. to assess whether KRT32 overexpression affects the studied pathways through NEMO or through alternative mechanisms rescue experiments are needed. Does knockdown/knockout of NEMO restore normal phenotype in KRT32-mutant cells? Does NEMO-knockdown/knockout result in observations that are comparable to those obtained with KRT32 -overexpression?

Reply: We made every effort to knock down NEMO in the Ker-CT cell line and primary keratinocyte cells using siRNA targeting NEMO mRNA. However, the transfection efficiency of the siRNA was too low to meet the experimental requirements. Additionally, the Ker-CT cell line already contains a puromycin marker gene, leaving fewer selection markers available for building cell lines overexpressing KRT32 WT and mutations using a hygromycin marker gene. As a result, we opted to use the NEMO inhibitor NBD as a substitute.

We used a well-recognized NEMO inhibitor, NBD (IKK γ /NEMO-binding domain), to reduce the function of NEMO in NF- κ B by disrupting its association with IKK β (M J May et al, Science, 2000, PMID: 10968790; Peterson et al, Molecular Medicine, 2011, PMID: 21267511). When the binding of NEMO and IKK β is inhibited by NBD, KRT32 mutations restore the level of IL-1 β secretion to the same level of KRT32 overexpression cells without drug treatment. And NBD treatment of the NC cells shows

a similar IL-1 β secretion to the KRT32 overexpressed cells in the context of NBD treatment. Together, all the data show that KRT32 is a key factor to affect the activation of NF- κ B pathway through the regulation of NEMO protein. New data has been included in **Figure 5I**.

Supplementary Figure 5I

(I) MSD assay detection of IL-1 β secretion in supernatant of Ker-CT cells overexpressing KRT32 wildtype and mutations using 10 μ M NBD-inhibitory peptide for 2 hours.

4. the reliance on HaCaT cells in all in vitro experiments is one of the major limitations of this work. HaCaT is a genetically abnormal, hypotetraploid cell lines with p53 and many other mutations. While findings from HaCaT cells can provide preliminary insights, making definitive conclusions about normal keratinocyte biology and disease mechanisms requires more physiologically relevant cell models. I recommend the authors address this limitation by performing key experiments such as: a) Assessing impact of KRT32 knockout/overexpression on NF- κ B activity and cytokine production in primary keratinocytes or immortalized keratinocytes (such as TERT-immortalized keratinocytes), b) Evaluating binding of KRT32 mutants to NEMO in primary or immortalized keratinocytes, c) Testing effects of KRT32 modulation on proliferation and inflammation in patient-derived PRP keratinocytes if available. In any case the authors need to discuss the limitations of the HaCaT model and interpreting results from these cells cautiously.

Reply: Thank you for your suggestion. We changed the HaCat cells by either immortalized Ker-CT (CRL-4048, ATCC) and primary human keratinocyte cells and re-performed the experiments accordingly in revised manuscript. We have re-performed all relevant experiments, including:

a) We assessed the impacts of KRT32 knockdown/overexpression on NF- κ B activity and cytokine production in Ker-CT and/or primary keratinocyte cells via the analysis of RNA-Seq, MSD, Western blot. All the results have been included in **Figure 4**.

b) The binding ability of KRT32 mutants to NEMO were analyzed by PLA immortalized Ker-CT cells in the revised **Figure 6D and 6E**, the detail method has been included in the Methods and Materials.

c) The skin samples of PRP patients with *KRT32* damaging mutations are from the

PFFE tissues, and we are unable to obtain enough fresh patient tissue since these patients has already recovered. Therefore, we used lentivirus-mediated stable expression of KRT32 WT and mutations and KRT32 knockdown in Ker-CT cells to analyze the effects of KRT32 modulation on the cellular proliferation and expression of inflammatory cytokines (**Figure 3 and Figure 4**).

5. The interaction between the NEMO and KRT32 was confirmed in HEK293T cells (in which KRT32 is not expressed under physiological conditions) and in cell-free systems. Considering the central place of this interaction in the model proposed by the authors, the interaction needs to be investigated using primary keratinocytes/epidermal lysates by co-IP. Can KRT32 pulled down by anti-NEMO Abs in keratinocytes?

Reply: We have redone these experiments in Ker-CT and primary keratinocytes accordingly, as follows:

1) We conducted GST-KRT32 pulldown assays using primary keratinocytes and identified the interacting protein NEMO through mass spectrometry and also validated the pulldown results by Western blot in both primary keratinocytes and Ker-CT cells (**Figure 5A - 5C**). 2) Co-IP experiments using anti-NEMO antibodies to pulldown KRT32 in both primary keratinocytes and Ker-CT cells further validates the interaction of KRT32 and NEMO (**Figure 5D and 5E**). 3) Proximity ligation assay (PLA) experiments were used to visualize and support this interaction, revealing a co-localization of KRT32 and NEMO in primary keratinocyte cells, Ker-CT cell line and normal skin tissue (**Figure 5F-5H**). Thus, all the current findings further validate the interaction between NEMO and KRT32 in keratinocytes, and the results has been included in the revised manuscript.

Figure 5 KRT32 interacts with NEMO

(A) Schematic diagram of the procedure to identify KRT32 interaction proteins in primary keratinocytes cells. *Escherichia coli* (*E. coli*) extracts containing GST (negative control) or GST-

KRT32 proteins were incubated with cell lysates of primary keratinocytes cells and glutathione-Sepharose beads to pull down GST complexes, and the bound proteins were then analyzed by mass spectrometry. (B, C) GST and GST-KRT32 proteins were incubated with lysates from primary keratinocytes cells and Ker-CT cells followed by Western blot analysis using anti-NEMO antibodies. (D, E) Co-IP analysis of the endogenous interaction of NEMO using anti-NEMO antibody in primary keratinocytes cells and Ker-CT cells. (F) PLA analysis of endogenous KRT32 and NEMO interaction in normal human adult skin. Scale bars represent 10 μ m. (G, H) PLA of endogenous KRT32 and NEMO in primary keratinocytes cells and Ker-CT cells. IgG was used as negative controls, and IKK α -NEMO interaction was used as positive control. Scale bars represent 10 μ m.

6. The authors claim that KRT32 promotes the K48-linked polyubiquitination-mediated degradation of NEMO protein and impedes the assembly of IKK complex (Figure 6). However, as far as this reviewer can tell, all experiments were performed in HEK293T cells. To validate the physiological relevance of KRT32-NEMO-interaction these results need to be confirmed in a keratinocyte-based system, especially considering that KRT32 does not seem to be expressed physiologically in HEK293T cells and its overexpression may result in non-physiological interactions.

Reply: Thanks for your comments. We have validated these results in Ker-CT cells accordingly, as follows:

- 1) The NEMO protein is prone to degradation via proteasomal pathway in the context of KRT32 overexpression (**Figure 7F and 7G**).
- 2) KRT32 WT increases the K48-linked ubiquitination of NEMO, but not the mutations. (**Figure 7J**)
- 3) PLA assays confirmed the influence of KRT32 WT and mutations on the ubiquitination of NEMO and IKK complex assembly (**Figure 7K-N**)

These experiments fully validated the physiological relevance of the KRT32-NEMO interaction and ubiquitination, and further strengthen the reliability our results.

Figure 7J-7L

(J) Immunoprecipitation of lysates with NEMO antibody from Ker-CT cells overexpressing KRT32 wildtype or mutations, and then detect the Ub-K48 modification of NEMO. (K) PLA assays in Ker-CT cells overexpression KRT32 wildtype and mutations to assess Ub interactions NEMO. Scale bars represent 10 μ m. (L) Quantification of the number of PLA foci (red) per nucleus detected in L. Each dot on the graph corresponds to a specific analyzed nucleus.

7. Description of the methods related to the treatment of KRT32 KO mice with TNF- α is missing. Information about the route, total number and frequency of administration, should be clearly described in the methods section, ideally with a graphical summary in Figure 8. Controls for the efficient treatment in WT and KO mice with TNF alpha are needed (i.e. the measurement of TNF-response genes before and after treatment).

Reply: Thank you for your suggestion. A graphical summary for the treatment of *KRT32* KO mice with TNF- α has been included in **Figure 8A** and as below, offering a clear depiction of the route, dosage, and frequency of TNF- α administration, and also described them in the Methods section.

To guarantee the TNF- α working in our experiments, we analyzed the transcription of TNF-response genes (*I11b*, *IL6* and *TNF α*) by qPCR, and validated the response to TNF- α in both WT and *Krt32* KO mice (**Figure 8F**, **Figure 8G** and **Supplementary Figure 8F**). And *Krt32* KO mice are more sensitive to TNF treatment to induce the inflammatory cytokines.

Figure 8A

(A) Schematic representation of the experimental schedule. Eight-week-old *Krt32* wildtype (WT) and knockout (KO) C57BL/6J mice (n=8 mice/group) were subcutaneously injected with TNF- α at a dose of 6 μ g/kg/day body weight into their shaved dorsal skin for a duration of 48 hours. The dorsal skin of *Krt32*^{-/-} mice exhibited pronounced thickening and extensive yellow scaling, resembling the cutaneous manifestations observed in human patients with pityriasis rubra pilaris (PRP). Skin, hair, and nail samples were collected for RNA-seq, H&E staining, cytokine assay, and scanning electron microscopy (SEM) analysis.

8. qPCR analysis was performed on three mice only, this is way too low to draw statistically valid conclusions considering the variability among mice. This number needs to be increased. On how many mice were the experiments presented on Figure 8A-D performed? Considering the variability among individual mice here, too the number should significantly more than two or three.

Reply: We have increased the number of mice to eight *Krt32* KO mice and eight matched WT mice (**Figure 8 and Supplementary Figure 8**). All the following experiments have conducted with all the eight pair of WT-KO matched mice, including the phenotype of the dorsal skin, H&E staining, immunohistochemistry of

Ki67, IL-1 β and NEMO, RNA-seq and followed by validation using qPCR. All the new results are consistent with our previous findings.

9. Additional experiments are needed are needed to evaluate the effect of Krt32 mutation on inflammation and NEMO-expression in the mouse epidermis. WB from epidermal cell lysates could be used to compare changes in NEMO between WT and KO mice. Which cells in the mouse epidermis express Krt32 protein? Is NEMO-expression altered in the corresponding cell layer in KO mice? Is the localization of p65 altered in the mutant mice?

Reply: The additional experiments to evaluate the effect of Krt32 mutation on inflammation and NEMO-expression in the mouse epidermis are shown as following:

1) we extracted the mouse epidermal tissue lysates and suckling mouse epidermal cell lysates. Both the two methods demonstrated higher expression of NEMO and phosphorylated p65 in *Krt32* KO mice detected by WB, suggesting the suppressing functions of KRT32 on the activation of NF- κ B signaling pathways by regulation of NEMO expression (**Figure 8K-8M**).

2) The mouse epidermis is very thin and only consists of 1-2 layer of cells in normal mouse skin, thus we referred to single-cell RNA sequencing of mouse skin and found the KRT32 also mainly expressed in basal keratinocytes (**refer to figure below**, single-cell sequencing source from four mouse samples GSM4547481, GSM4547482, GSM5024746, GSM5024747, GSM5795802; PMID: 38472183). Further, we also tried our best to do IHC of NEMO in *Krt32* KO mice, the NEMO expression mainly expression in the supra-basal layer, and loss of KRT32 leads to the expression of NEMO in both the basal and supra-basal layers, which are consistent with the result of *KRT32* damaging mutation patients (**Figure 8J and Supplementary Figure 8E**).

KRT32 tissue expression based on the datasets of single-cell RNA sequencing of mouse skin

2) We have showed the overexpression of p-p65 in skin of *Krt32* KO mice by western blot (**Figure 8K**), furthermore, we also performed IHC using p65 antibody and found p65 expression and ratio nuclear p65 positive cells increased in basal layers (**refer to figure below**). Thus, our mouse model is consistent with the PRP patients and demonstrates the suppressing role of KRT32 on NF- κ B activation.

(A) Representative immunohistochemical staining of the dorsal skin of *Krt32* WT and KO mice treated with TNF- α for detecting p65. (B) Statistical analysis nuclear p65 positive cells in basal layers (n = 6).

10. Transcriptomic analysis of TNF- α -stimulated HaCat cells with overexpression of KRT32 was performed, however results seem to be shown only for four transcripts. The results from these experiments should be provided in the form of supplementary tables along with false-discovery-rate adjusted p-values and fold changes. Was the IL-1 pathway significantly enriched among the differentially expressed genes? What was the results of unbiased analysis, such as GO-enrichment analysis on this data set? Was NF κ B-pathway affected? Were there any other processes altered, too? Results from a more detailed analysis need to be presented and discussed. Why did the authors focus on IL1A/B, and IL1Rs in this analysis?

Reply: We performed RNA-sequencing and transcriptomic analysis of different samples, including PRP patient skin tissue, Ker-CT cells with KRT32 overexpression, and *Krt32* KO mice skin tissue with their respective controls, and all the results along with false-discovery-rate adjusted p-values and fold changes are now available in **Supplementary Figure 3A and 3B, Figure 4A and 4B, Figure 8D and 8E, and Supplemental Table 5-8, Supplemental Table 10 and 11.**

From the KEGG analysis, TNF and NF- κ B pathways rank the top two in samples from PRP patients, mouse and cell models. We also found that the downstream effector genes of NF- κ B are also regulated by the KRT32, including TNF, IL1A, IL1B, IL6 and others (the revised **Figure 4A and 4B**). Except for that, the JAK-STAT and Sphingolipid signaling pathway was also found to be enriched in all samples (**refer to Figures with comment #1**).

As for IL-1 pathway, the transcription of relative genes shows a significant difference between HaCat and *Krt32* KO mouse and PRP patients. In HaCat cells, IL1A, IL1B, IL1RN and IL1R2 are among the top genes in union set of gene lists in the NF- κ B pathway. Differently, in PRP patients and *Krt32* knockout mice, the transcription

IL1RN shows decrease and IL1R2 has no significantly difference (refer to **Figure 11**). Previous publications reported that P53 mutant could affect the expression of immune cytokine and repress the expression of IL1RN (PMID: 37602328, PMID: 24998848), probably attributed to the genetic deficient background of HaCat cell line. Due to the inconsistency, we decided to remove this part from manuscript to make sure the accuracy of our study. Thank you again for point out the defect of HaCat, it would be not our choice for the research of skin immunity.

Quantification of IL1RN (A) and IL1R2 (B) mRNA levels in HaCat cells, Ker-CT cells overexpressing or knocked out for KRT32, as well as in PRP patients with *KRT32* mutations and *Krt32* KO mice using RNA-seq. Data from Supplementary table 6, 8 and 11.

11. The analysis of inflammatory cytokine levels in PRP skin was performed by IHC, which is a semi-quantitative method. How were the results quantified? Data obtained with quantitative method needs to be provided (e.g. RNAseq, qRT-PCR).

Reply: To provide a quantitative assessment, we conducted RNA-seq on paraffin-embedded tissue sections from PRP patients and their matched healthy controls (6 PRP patients with *KRT32* mutation vs. 6 healthy controls), and samples from PRP mouse model (8 *Krt32* KO mice vs. 8 *Krt32* WT mice). The transcriptional profile data is now present in **Supplemental table 6, 8 and 11**. The relevant gene transcriptions of inflammatory cytokines from these transcriptomes are shown in **Figure 4A and 4B** accordingly. And qPCR was used to validate the transcriptome data of PRP mouse skin samples after TNF- α treatment (**Figure 8F and 8G**).

12. The figure legend for Supplementary Figure 4D seems to be incorrect. “Immunohistochemistry of KRT32 expression in skin tissues obtained from PRP patients with KRT32 mutations, PRP patients without KRT32 mutations, and healthy individuals” -however IF images are shown only for healthy individuals and patients with KRT32 mutations.

Reply: We sincerely apologize for our oversight. The new IF images have been included in the revised **Supplementary Figure 1D**. To consolidate the conclusion, we re-performed this experiment, and included all the samples from 6 PRP patients with/without *KRT32* mutations and the matched 6 healthy controls.

13. The authors claim that direct immunofluorescence assays demonstrated colocalization of KRT32 and NEMO in HaCaT cells (Figure 4). However, these are low-magnification IF-images which are not conclusive in terms of co-localization. Proximity ligation assay, or super-resolution microscopy would be need with quantitative image analysis to support this claim. Again, this result was obtained from HaCaT cells. Are KRT32 and NEMO colocalized in primary keratinocytes?

Reply: PLA to show the colocalization of KRT32 and NEMO in human skin tissue, primary keratinocytes and Ker-CT cells, the relevant results is now presented in the **Figure 5F-5H**.

Figure 5F-5H

(F) PLA analysis of endogenous KRT32 and NEMO interaction in normal human adult skin. Scale bars represent 10 μm. (G, H) PLA of endogenous KRT32 and NEMO in primary keratinocytes cells and Ker-CT cells. IgG was used as negative controls, and IKK α -NEMO interaction was used as positive control. Scale bars represent 10 μm.

14. Figure 3C: it is surprising that treatment of keratinocytes with TNF does not result in increased p65-phosphorylation – Is this a technicality due to the use of HaCaT cells or is this an experimental variation? It seems to contradict to panel E, where nuclear p65 is increased at the same time point.

Reply: We re-performed the experiments using the Ker-CT and primary keratinocyte cells to replace HaCat cells based on your comment #4. Knockdown of KRT32 in Ker-CT cells increases the p65-phosphorylation compared to NC (**Figure 4D**), and KRT32 OE in Ker-CT and primary keratinocyte show a decrease of the p65-phosphorylation signal intensity (**Figure 4H and 4I**), which are consistent with the results from HaCat cells. Subsequently, we further overexpressed the PRP damaging *KRT32* mutations in Ker-CT cells and found a increased p65-phosphorylation signal intensity compared to *KRT32* WT OE (**Figure 4M**). All the results validate the roles of *KRT32* in the suppression of p65-phosphorylation.

As for the original Figure 3C, to avoid the oversaturation of the p-P65 signal in Lane 4 (KO+TNF- α), we shortened the exposing time when developing this membrane, which leads to a weak signal in Lane 3 (WT+TNF- α) and still strong signal in Lane 4 (KO+TNF- α), but we still see a slight increase signal intensity in Lane 3 (WT+TNF- α) compared to Lane 1 (WT-TNF- α). As for the Panel E, we isolated the nuclei from the cells and used pan p65 antibody to do test, we also see a increase of nuclear p65 in KO+TNF- α compared to WT+ TNF- α in both the two incubating time. The difference of signal intensity between Panel C and E might arise from the different protein isolation methods (total cell lysis vs. nuclear isolation) and/or the different antibodies (p-p65 vs. pan-p65 antibodies).

Figure 4D-4I

(D) The phosphorylated p65 expression in KRT32 knockdown Ker-CT cells with and without TNF- α (20 ng/mL) treatment for the indicated time. (E-G) Detection of IL-1 β , IL-6, IL-8 secretion in supernatant of KRT32 knockdown Ker-CT cells with or without TNF- α treatment by MSD assay. (H-I) The phosphorylated p65 expression in primary keratinocytes and Ker-CT cells with Flag-tagged KRT32 overexpression.

15. Figure 7F: western blot data should be shown about the expression of KRT32 in

these conditions. I find it very surprising that overexpression of K32 could completely block IL-1 β -induced p65-phosphorylation. What is the authors explanation for this? What is the baseline Krt32-expression in these cells? IL1R signals through several pathways, can Krt32 inhibit all of them?

Reply: Due to the inconsistency of IL1RN and IL1R2 between in HaCat and in Ker-CT, PRP patient tissue and mouse model (**refer to Figures with comment #10**), the IL1RN and IL1R2 transcriptional changes of IL1 signaling pathway might be specific for HaCat cell line. To ensure the scientific rigor of this study, we decided to remove this part from our revised manuscript.

Previous publications reported that P53 mutant could affect the expression of immune cytokine and repress the expression of IL1RN (PMID: 37602328, PMID: 24998848). It is plausible that the observed complete inhibition of IL-1 β -induced p65-phosphorylation by KRT32 overexpression might be related to p53 or other unknown genetic changes in HaCat cells.

16. I wonder whether alterations in cell—to-cell or cell-to-ECM junctions are affected in KRT32-mutant patients and whether signalling from the junctional complexes (e.g. via catenins) could mediate the observed effects. Does loss-of-function of KRT32, similar to that of other intermediate filaments, impact desmosomal, hemi-desmosomal interactions? Did the authors see any signs of this in their histological and EM- analyses? Please discuss this and if appropriate provide additional images covering areas of the basal lamina and cell junctions in Supplementary Figure 7.

Reply: Given that all the six KRT32 mutant PRP patients have recovered, we cannot get the fresh lesional skin tissues in the process of illness process. Instead, we collected skin samples from our PRP mouse model to analyze through electron microscopy. We took 5000 x and 30000 x amplified images, and analyzed the structural integrity of keratinocyte. In *Krt32* KO mice, neither the desmosomes (arrows in red) nor the hemidesmosomes (arrows in yellow) manifested significant alterations. Besides, we did not observe the abnormal keratinocyte junctional structures in H&E staining images of PRP patients with *KRT32* mutations (**Figure 1**). Thus, we supposed that KRT32 may not directly participate in the formation of cell junctions in skin, but regulate skin immune homeostasis as we described in our revised manuscript. All the new data has been included in the revised **supplemental Figure 10**.

Supplementary Figure 10. The ultrastructure of keratinocytes in a model of PRP-like dermatitis *Krt32*^{-/-} mice observed by electron microscopy.

(A) The overall keratinocyte cell phenotypes in *Krt32*^{-/-} and wildtype mice observed with and without TNF treatment by electron microscopy. Scale bar = 5 μm. (B, C) The representative images of hemidesmosomes and desmosomes of keratinocytes from PRP mouse model. Scale bar = 1 μm. Hemidesmosomes (arrows in yellow); Desmosomes (arrows in red) (D) The number of hemidesmosomes per μm. At least 13 fields for each sample from different mice were quantified. (E) The length of individual desmosome, >100 desmosomes were quantified for each sample from different mice. Data are means ± SEM, and P value was calculated using a two-sided unpaired Student's t test (ns: no significant difference).

17. How was the specificity of the antibodies used for the immunohistochemistry analysis of IL-1R2 and IL-1Ra confirmed? The images provided show mainly cytoplasmic and, in some cases, even nuclear expression for them. is this the expected staining patter for these proteins in normal epidermis?

Reply: Thank you to point this. They are very interesting phenomena for the subcellular localization of IL-1R2 and IL-1Ra. Here we used the antibodies from Abcam (ab212208) and PROTEINTECH (10844-1-AP), and the specificity of antibodies was checked by WB. The relevant images are shown as below:

Immunoblot analysis of IL1-Ra and IL1-R2 expression in HaCat and Ker-CT cells lysates.

Regarding IL-1Ra, we did observe some nuclear expression. Upon reviewing multiple literatures, we found that IL-1Ra has different isoforms, with isoforms 1-3 being expressed in the cytoplasm and/or nucleus (see PMID: 15923123; PMID: 37025835). Additionally, according to Protein Atlas data, IL-1Ra is expressed in both the cytoplasm and nuclei of esophagus, skin, and tonsil tissues detected by IHC (refer to <https://www.proteinatlas.org/ENSG00000136689-IL1RN/tissue/skin#>, where the skin samples refer to SKIN 2 - Antibody staining- Antibody CAB009633).

Interestingly, IL-1Ra expression varies across different cell lines. For instance, in A-431 cells, IL-1Ra is mainly expressed in the cytosol, while in the RT-4 cell line, it is expressed in both the cytoplasm and nuclei (see <https://www.proteinatlas.org/ENSG00000136689-IL1RN/subcellular#human>, where the antibody used was from Sigma-Aldrich (HPA001482)). We also checked more normal skin tissue samples and found that the majority of IL-1Ra is expressed in the cytoplasm. Therefore, we hypothesize that nuclear IL-1Ra expression may depend on specific isoform expression, and different expression patterns of IL-1Ra isoforms may occur under specific conditions.

Immunohistochemical staining was utilized to analyze the expression levels of IL1Ra in skin tissue as well as A-431 and RT-4 cell lines. The figures presented herein were derived from Protein Atlas data.

Regarding IL-1R2, upon closer examination of all the images of IHC staining, we observed that the staining phenotype does not exhibit a clear nuclear outline (**see figure below**). The compact nature of keratinocytes in human skin may give the impression that IL-1R2 is localizing to the nuclei. Therefore, based on our findings, we believe that IL-1R2 still localizes in the cytoplasm or on the cellular membrane.

Immunohistochemical staining was utilized to analyze the expression levels of IL-1R2 in skin tissue

In light of the inconsistencies observed in the expression changes of IL-1R2 and IL-1RA in the HaCat cell line compared to those in Ker-CT cells, PRP patients, and the mouse model, as mentioned in comment #15, we have decided to remove this section from the revised manuscript to ensure the accuracy of our conclusions.

18. The heritability of PRP is non-mendelian, thus heterozygous KRT32 mutations are unlikely to be the sole reason for the disease. The authors should discuss other factors that would be needed to trigger disease onset in combination with KRT32 mutations.

Reply: We appreciate your suggestion and agree that these *KRT32* mutations alone are unlikely to be the sole reason for the disease and external environmental stimuli are needed to collaborate with these mutations to trigger the onset of the disease. We have described this point in the Result part and included the detailed discussion of *KRT32* location in Discussion part (as below) (Lines 406-417).

*“It has been reported that microbial infection may act as a predisposing factor for PRP, leading to the upregulation of TNF- α expression and activation of skin inflammation⁴²⁻⁴⁷. In our study, we observed high levels of TNF- α in both serum and skin lesions of PRP patients. The healed PRP patient exhibits much lower serum TNF level compared to the serum TNF level in patients before treatment. And only following subcutaneous administration of TNF- α induced PRP-like manifestation in the dorsal skin of *Krt32*^(-/-) mice, suggesting a potential role for TNF- α in the pathogenesis of PRP. Meanwhile, the intrinsic *Tnf* transcription of keratinocyte cells in PRP mouse model increased in the process of TNF treatment compared to that before treatment and after restored (Supplementary Figure 8F). Therefore, *Krt32* knockout may predispose individuals to PRP rather than spontaneous skin inflammation directly.”*

REVIEWERS' COMMENTS

Reviewer #1 (Remarks to the Author):

The authors properly addressed most of the comments this reviewer raised.

Reviewer #2 (Remarks to the Author):

The revisions performed by the authors have enhanced the quality of the manuscript.

However, one minor point should be addressed.

The comments regarding the role of K32 mutations in promoting PRP are clearer.

1. Item 3, the histologic features described by both of your pathologists are accurate but incomplete. Focal intraepidermal acantholysis has been described in human biopsies of PRP.

Please comment on this, in the images provided I did not see acantholysis.

Please review manuscript PMID: 30203619 and others. If not present, this type of PRP may be unique and not be associated with epidermal acantholysis.

Reviewer #3 (Remarks to the Author):

The authors have answered my questions.

REVIEWERS' COMMENTS

Reviewer #1 (Remarks to the Author):

The authors properly addressed most of the comments this reviewer raised.

Reviewer #2 (Remarks to the Author):

The revisions performed by the authors have enhanced the quality of the manuscript. However, one minor point should be addressed.

The comments regarding the role of K32 mutations in promoting PRP are clearer.

1. Item 3, the histologic features described by both of your pathologists are accurate but incomplete. Focal intraepidermal acantholysis has been described in human biopsies of PRP. Please comment on this, in the images provided I did not see acantholysis. Please review manuscript PMID: 30203619 and others. If not present, this type of PRP may be unique and not be associated with epidermal acantholysis.

Reply: Thank you for your insightful comment. We reviewed relevant literatures and agree with you that focal intraepidermal acantholysis could be found in some human PRP biopsies. However, upon carefully re-examining all the H&E staining images of our PRP patients, acantholysis was not observed just like some other reported PRP cases (PMID: 22703878; 35529074; 19946540; 37704911). Thus, we think that this phenomenon may occur only in specific types of PRP. We hope to have the opportunity to further elucidate this in the future.

Reviewer #3 (Remarks to the Author):

The authors have answered my questions.